# Regional severe particle pollution and its association with synoptic weather patterns in the Yangtze River Delta region, China

**Lei Shu [1], Min Xie [1*], Da Gao [1], Tijian Wang [1*], Dexian Fang [2], Qian Liu [3], Anning Huang [1], Liwen Peng [1]**

[1] School of Atmospheric Sciences, CMA-NJU Joint Laboratory for Climate Prediction Studies, Jiangsu Collaborative Innovation Center for Climate Change, Nanjing University, Nanjing 210023, China

[2] Chongqing Institute of Meteorology and Science, Chongqing 401147, China

[3] Jiangsu Provincial Academy of Environmental Science, Nanjing 210036, China

-----------------------------------------------------------------------------------------------------

∗Corresponding to Min Xie (minxie@nju.edu.cn) and Tijian Wang (tjwang@nju.edu.cn)

**Abstract:** Regional air pollution is significantly associated with dominant weather systems. In this study, the relationship between the particle pollution over the Yangtze River Delta (YRD) region and weather patterns is investigated. First, the pollution characteristics of particles in the YRD are studied using in situ monitoring data ($PM_{2.5}$ and $PM_{10}$) in 16 cities and Terra/MODIS AOD (aerosol optical depth) products collected from December 2013 to November 2014. The results show that the regional mean value of AOD is high in the YRD, with an annual mean value of 0.71±0.57. The annual mean particle concentrations in the cities of Jiangsu Province all exceed the national air quality standard. The pollution level is higher in inland areas, and the highest concentrations of $PM_{2.5}$ and $PM_{10}$ are 79 and 130 $\mu g \cdot m^{-3}$, respectively, in Nanjing. The $PM_{2.5}/PM_{10}$ ratios are typically high, thus indicating that $PM_{2.5}$ is the overwhelmingly dominant particle pollutant in the YRD. The wintertime peak of particle concentrations is tightly linked to the increased emissions during the heating season, as well as adverse meteorological conditions. Second, based on NCEP reanalysis data, synoptic weather classification is conducted, and five typical synoptic patterns are objectively identified. Finally, the synthetic analysis of meteorological fields and backward trajectories are applied to further clarify how these patterns impact particle concentrations. It is demonstrated that air pollution is more or less influenced by

high-pressure systems. The relative position of the YRD to the anti-cyclonic circulation exerts
significant effects on the air quality of the YRD. The YRD is largely influenced by polluted air
masses from the northern and the southern inland areas when it is located at the rear of the East
Asian major trough. The significant downward motion of air masses results in stable weather
conditions, thereby hindering the diffusion of air pollutants. Thus, this pattern is quite favorable
for the accumulation of pollutants in the YRD, resulting in higher regional mean $PM_{10}$
(116.5±66.9 $\mu g \cdot m^{-3}$), $PM_{2.5}$ (75.9±49.9 $\mu g \cdot m^{-3}$) and AOD (0.74) values. Moreover, this pattern is
also responsible for the occurrence of most large-scale regional $PM_{2.5}$ (70.4%) and $PM_{10}$ (78.3%)
pollution episodes. High wind speed and clean marine air masses may also play important roles in
the mitigation of pollution in the YRD. Especially when the clean marine air masses account for a
large proportion of all trajectories (i.e., when the YRD is affected by the cyclonic system or
oceanic circulation), the air in the YRD has a smaller chance of being polluted. The observed
correlation between weather patterns and particle pollution can provide valuable insight into
making decisions about pollution control and mitigation strategies.
**Keywords:** $PM_{2.5}$; $PM_{10}$; air pollution meteorology; synoptic weather pattern; the Yangtze River
Delta region

**1. Introduction**
The common occurrence of regional particle pollution has acquired worldwide attention in
the scientific community (Malm et al., 1994; Putaud et al., 2004; Chan and Yao, 2008) due to its
adverse impacts on visibility (Singh and Dey, 2012; Green et al., 2012) and public health (Kappos
et al., 2004; Brook et al., 2010). Generally, the causes of this kind of pollution involve diverse
aspects. Two major contributors to this pollution include the emission of pollutants and weather
conditions (Oanh and Leelasakultum, 2011; Young et al., 2016). Particle pollution in urban
agglomerations is primarily attributed to very large amounts of the anthropogenic emissions of
primary particles and their precursors (e.g., $SO_2$, $NO_x$, VOCs). However, these emissions are
normally quasi-stable within a certain period of time (Kurokawa et al., 2013). Thus, the pollution
level in a certain region generally depends on the regional weather conditions (namely, weather
patterns), which are strongly correlated with synoptic-scale atmospheric circulation (Buchanan et
al., 2002; Chuang et al., 2008; Flocas et al., 2009; Zhang et al., 2012; Zhao et al., 2013; Russo et
al., 2014; Grundstrom et al., 2015; Zheng et al., 2015a; 2015b; Li et al., 2016).

To date, researchers have gained an improved knowledge of the relationship between weather

patterns and particle pollution. For example, Buchanan et al. (2002) observed significantly
elevated concentrations of Black Smoke and $PM_{10}$ under the anti-cyclonic, southerly and
southeasterly weather types in the city of Edinburgh in the UK between 1981 and 1996. Russo et
al. (2014) presented an objective classification scheme for the atmospheric circulation affecting
Portugal between 2002 and 2010 and revealed that higher concentrations of $PM_{10}$, $O_3$ and $NO_2$ are
predominantly associated with synoptic circulation that is characterized by an eastern component
and the advection of dry air masses. Previous studies have confirmed that different levels of air
pollution are closely related with weather patterns, and they ascribed its great spatial variability to
the fact that the dominant weather pattern differs between different regions (Flocas et al., 2009;
Grundstrom et al., 2015).

In recent decades, the air pollution caused by $PM_{10}$ and $PM_{2.5}$ has become an extremely

prominent air quality problem in the urban areas of China (Deng et al., 2011; Huang et al., 2012;
Ji et al., 2012; Cheng et al., 2013; Kang et al., 2013; Huang et., 2014; Zhang et al., 2014; Xie et al.,
2016a; 2016c; Zhu et al., 2017). Many studies have tried to reveal the meteorological
contributions to these severe particle pollution episodes. Chuang et al. (2008) identified seven
weather patterns for aerosol events occurring from March 2002 to February 2005 in the Taipei
Basin and suggested that weather systems and their associated terrain blocking played important
roles in the accumulation of $PM_{2.5}$ during the days of events. Niu et al. (2010) revealed the
potential impacts of the weakening of the East Asian monsoon circulation and increased aerosol
loading on the increase in wintertime fog in China. Zhao et al. (2013) analyzed a regional haze
episode in the North China Plain from 16 to 19 January 2010 and noted that strong temperature
inversion, weak surface wind speed and descending air motions in the boundary layer were
responsible for the accumulation of pollutants in a shallow layer that produced high pollutant
concentrations within the source region. Zheng et al. (2015a) found that favorable atmospheric
circulation conditions are responsible for the severe winter haze over northeastern China. Li et al.
(2016) noted that the fog-haze days over central and eastern China exhibited the clear features of
inter-annual variations and that the strong (weak) East Asian winter monsoon may result in less
(more) fog-haze days throughout this region.

The Yangtze River Delta (YRD) region, which is located in the southeastern coastal area of

East China, is one of the most developed urban economic regions in the world; it generally
includes Shanghai, Jiangsu Province and Zhejiang Province, and it occupies over 20% of China's
total gross domestic product (GDP) (Shu et al., 2016; Xie et al., 2016a; 2017). In recent years,
similar to other megacity clusters in China, such as the Beijing-Tianjin-Hebei (BTH) region (He et
al., 2001; Chan and Yao, 2008; Ji et al., 2012; Zhang et al., 2012; 2014; Zhao et al., 2013; Zheng
et al., 2015a) and the Pearl River Delta (PRD) region (Ho et al., 2003; Chan and Yao, 2008; Xie et
al., 2016c; Zhu et al., 2017), the YRD has suffered from severe air pollution problems caused by
an increasing population, urban expansion, and industrialization (Chan and Yao, 2008; Fu et al.,
2008; 2010; 2014; Deng et al., 2011; Li et al., 2011; Huang et al., 2012; Kang et al., 2013; Wang et
al., 2013; 2014; 2015; Xie et al., 2014; 2016a, 2016b, 2017; Feng et al., 2015; Zheng et al., 2015b;
Shu et al., 2016; Xu et al., 2016; Ming et al., 2017). In particular, severe particle pollution
episodes are widely recognized as one of the major air pollution issues in the YRD (Fu et al., 2008;
2010; Deng et al., 2011; Huang et al., 2012; Kang et al., 2013; Kong et al., 2013; Wang et al.,
2013; 2014; 2015; Fu et al., 2014; Feng et al., 2015; Zheng et al., 2015b; Xu et al., 2016; Ming et
al., 2017). Thus, many studies have been conducted to determine the contamination status (Fu et
al., 2010; Kang et al., 2013; Wang et al., 2013; 2015; Feng et al., 2015; Ming et al., 2017),
possible source (Fu et al., 2010; 2014; Kong et al., 2013; Wang et al., 2013; 2014; Xu et al., 2016),
and causes or features (Fu et al., 2008; 2010; Huang et al., 2012; Wang et al., 2015; Zheng et al.,
2015a) of these episodes. However, studies that have attempted to determine how particle
pollution in the YRD is associated with synoptic weather patterns are still quite limited. Zheng et
al. (2015b) summarized the synoptic-scale atmospheric circulations influencing the distribution of
particles over eastern China during autumn from 2001 to 2010. They found that there are six
polluted weather types and three clean ones and revealed that heavy pollution events most
commonly occur when the study areas are located at the rear of the anticyclone. However, their
study considered the influence of pollution in a region that is larger than YRD, only focused on
pollution in October, and was mainly based on satellite aerosol optical depth (AOD) data.
Ground-based monitoring particle concentration data can better represent the status of particle

pollution in the urban atmosphere of the YRD. Thus, to better understand the relationship between pollution in the planetary boundary layer and the synoptic weather patterns over the YRD, further studies should be conducted based on surface monitoring data collected over a time period of at least one year in the YRD.

This work attempts to enhance our understanding of particle pollution in the YRD and provide scientific knowledge about the association of regional severe particle pollution and synoptic weather patterns. First, we analyze the spatial and temporal distribution of $PM_{10}$, $PM_{2.5}$ and AOD in the YRD from December 2013 to November 2014 to illustrate the characteristics of particle pollution over this region. Second, synoptic weather classification is conducted to reveal the weather patterns related to heavy pollution. Finally, the synthetic analyses of meteorological fields and backward trajectories are used to further clarify the impact mechanism. In this paper, Section 2 describes the observed data, the synoptic weather classification method and the trajectory model. Section 3 presents our main findings, including a detailed analysis of the characteristics of particle pollution in the YRD, the synoptic weather patterns affecting this pollution, and the mechanism by which weather systems impact pollution. Finally, a brief summary is presented in Section 4.

## 2. Data and methods

### 2.1 Observed data

The observed air quality data used in this study are obtained from the National Environmental Monitoring Center (NEMC) of China. The in situ monitoring data of the hourly concentrations of $PM_{2.5}$, $PM_{10}$, CO, $NO_2$, $SO_2$ and $O_3$ are acquired from the national air quality real-time publishing platform (http://106.37.208.233:20035). Sixteen cities are selected as representative research sites to better reflect the status of particle pollution over the YRD region. These cities include Shanghai, Changzhou, Nanjing, Nantong, Suzhou, Taizhoushi, Wuxi, Yangzhou, Zhenjiang, Hangzhou, Huzhou, Jiaxing, Ningbo, Shaoxing, Taizhou, and Zhoushan (here, Taizhou in Jiangsu Province is referred to as Taizhoushi to distinguish it from the city of Taizhou in Zhejiang Province). Fig. 1 shows the locations of the 16 cities in the YRD. In order to better characterize the pollution levels of each city, the hourly pollutant concentration of each city is calculated as the average value of the pollutant concentrations measured in several of the

national monitoring sites in that city,. The sampling methods and the quality assurance and quality
control (QA/QC) procedures used at each site are in accordance with the Chinese national
standard HJ/T193-2005 (State Environmental Protection Administration of China, 2006; Xie et al.,
2016b). Furthermore, manual inspection is conducted during data processing; this inspection
includes the removal of missing and abnormal values (e.g., $PM_{2.5}$ values that are higher than $PM_{10}$
values). The study period lasts from December 2013 to November 2014. In the following analysis,
winter refers to the period from December 2013 to February 2014. Accordingly, spring, summer
and fall represent the periods from March to May, June to August, and September to November
2014, respectively.

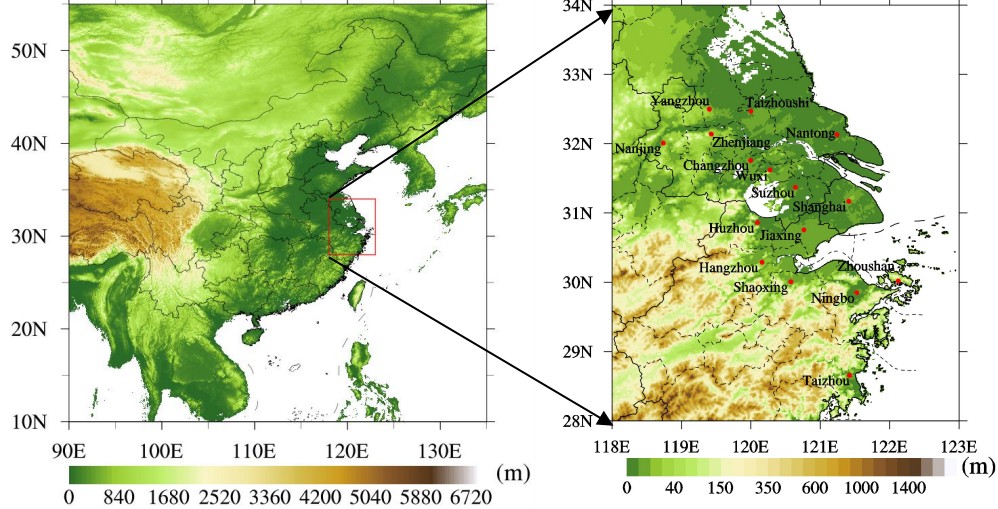

**Figure 1. The location of the YRD in China (a) and 16 typical cities in the YRD (b), with terrain elevation**
**data.      The       terrain        elevation       data       are      obtained       from       the        website**
**(https://www.ngdc.noaa.gov/mgg/global/relief/ETOPO1/data/bedrock/cell_registered/).**

The use of Moderate Resolution Imaging Spectroradiometer (MODIS) aerosol products can
help us comprehensively analyze the spatial and temporal variations in aerosol loading over China.
In this study, we use the aerosol optical depth (AOD) data obtained at a wavelength of 550 nm in
the Terra/MODIS daily global Level 3 products (MOD08_D3). These data can be obtained from
the   MODIS   collection   6   (C6)   dataset   (https://ladsweb.nascom.nasa.gov/search/index.html).
MODIS aerosol products are derived using two entirely independent retrieval algorithms: one is
used for deriving aerosols over land (Chu et al, 2002; 2003) and another is used for deriving
aerosols over the ocean (Remer et al, 2002; 2005; Chu et al., 2005). Here, we use the C6 Deep
Blue (DB) products to derive aerosols over land, with a spatial resolution of 1° × 1°, during the
period from December 2013 to November 2014. For detailed descriptions of the retrieval
algorithms and their accuracy and validation, refer to the work of Hsu et al. (2013).
To illustrate actual weather situations, the hourly monitored meteorological parameter
records in each of the 16 typical cities are also applied. These data include 2 m temperature (T), 2
m relative humidity (RH), 10 m wind speed (WS), 10 m wind direction (WD) and surface air
pressure (P). These data are collected from the National Meteorological Center
(http://www.nmc.cn).

**2.2 Synoptic weather classification**
Synoptic weather classification refers to the analysis of historical weather charts and the
characterization of weather systems. It is more effective for producing disastrous weather forecasts
due to its ability to reveal atmospheric circulation situations. With the gradual popularization of
computer analysis and the increased sharing of data, synoptic weather classification has great
practical value in a wide variety of research fields. For example, it has widespread applications in
the field of analyzing weather patterns related to air pollution (Mcgregor and Bamzelis, 1995;
Zhang et al., 2012; Santurtún et al., 2015).
Methods of synoptic weather classification can generally be divided into objective and
subjective methods (El-Kadi and Simithson, 1992). In this study, we apply the sums-of-squares
technique, which is an objective classification method that was established in 1973 by Kirchhofer
(Kirchhofer, 1973). The sums-of-squares technique can effectively categorize more than 90% of
analyzed weather maps, which represents an improvement over other correlation techniques
(Yarnal, 1984). The application of this technique involves three steps. First, the daily pressure data
at each grid point are normalized as follows:

$$Z_i = \frac{(X_i - \overline{X})}{s} \tag{1}$$


where $Z_i$ is the normalized value of grid point i, $X_i$ is the value at grid point i, $\overline{X}$ is the mean
value of the study domain, and s is the standard deviation. Data normalization removes the effects
of the magnitude of pressure and improves the seasonal comparability of different weather types.
Second, each normalized grid point is compared to all other grid points based on the Kirchhofer
score (S) of each grid point:
$$S = \sum_{i=1}^{N} (Z_{ai} - Z_{bi}) \tag{2}$$

where $Z_{ai}$ is the normalized value of grid point i on day a, $Z_{bi}$ is the normalized value of grid point
i on day b, and N is the number of grid points. The Kirchhofer score (S) is calculated for each row
(denoted as $S_R$), each column ($S_C$) and the entire study domain ($S_T$) to ensure the pattern similarity
between any pair of patterns for all grid points. Finally, all days are separated into one of the
identified synoptic weather patterns based on these three values and their empirically derived
thresholds. Thus, the values of $S_R$, $S_C$ and $S_T$ must be lower than their respective threshold values
for these patterns to be accepted as similar (Barry et al., 1981). For each daily grid, the lowest
significant Kirchhofer score (S) is recorded with the associated key day, thus denoting the
synoptic type of that day. All remaining days are considered to be 'unclassified'.

The meteorological field dataset used in the sums-of-squares technique contains NCEP–DOE

AMIP-II Reanalysis 2 data (Kanamitsu et al., 2002), which are collected at 00:00, 06:00, 12:00,
and        18:00        UTC        (universal        time        coordinated)
(https://www.esrl.noaa.gov/psd/data/gridded/data.ncep.reanalysis2.pressure.html). These data have
144×73 horizontal grids with a grid spacing of 2.5°. From the ground level to 10 hPa, there are 17
pressure levels in the vertical direction. The classification of synoptic weather maps is conducted
using the gridded data at a geopotential height of 850 hPa during the same time period when the
air quality data are recorded. The domain of interest is centered over the YRD region, covering an
area of 25-40° N in latitude and 110-128°E in longitude.

**2.3 HYSPLIT model**

Backward trajectories can be adopted to help understand transport paths and identify the

source regions of air masses. The Hybrid Single-Particle Lagrangian Integrated Trajectory
(HYSPLIT) Model (Version 4) was developed by the National Oceanic and Atmospheric
Administration (NOAA) Air Resources Laboratory (ARL). It is one of the most extensively used
atmospheric transport and dispersion models for the study of air parcel trajectories (Draxler and
Rolph, 2013; Rolph, 2013; Stein et al., 2016), and it has been widely applied in simulations of the
complex transport, diffusion, chemical transformation and deposition processes of atmospheric
pollutants (Mcgowan and Clark, 2008; Wang et al., 2011; Huang et al., 2015; Xie et al., 2016b).
In this study, HYSPLIT is used to compute the backward trajectories of air parcels, reveal the
possible source regions of air masses, and establish source-receptor relationships for each synoptic
weather pattern. For each synoptic weather pattern, the terminus of each trajectory is considered to
be located at the observation site in Nanjing (32°N, 118.8°E). The 72-h backward trajectories are
then calculated and clustered. The ending point is defined as 1500 m above sea level. The NCEP
reanalysis data (http://ready.arl.noaa.gov/archives.php) are used to drive the backward trajectory
calculation. The NCEP data contain 6-hourly basic meteorological fields on pressure surfaces with
a spatial resolution of 2.5°. In this study, these data are also converted to hemispheric 144 by 73
polar stereographic grids; these data thus have the same grid configuration as the dataset applied
in the synoptic weather classification.

**3. Results and discussion**
**3.1 Characteristics of particle pollution in the YRD**
**3.1.1 Spatial distributions of particle pollution**
Fig. 2a displays the annual mean values of AOD observed at a wavelength of 550 nm
throughout most of China. The highest values (i.e., larger than 0.6) generally occur in the BTH,
the YRD, the Sichuan Basin (SCB), and some of the central and southern provinces in China (i.e.,
Hubei, Hunan and Guangxi provinces). AOD is mainly governed by fine particles in industrialized
urban conditions (Kim et al., 2006); thus, the abovementioned areas should suffer from high
columnar aerosol loading. In the YRD, with the development of modern industrialization and
urbanization, contrasts in the atmospheric pollution levels among different cities gradually
decrease, and severe air pollution episodes tend to exhibit significant regional pollution
characteristics.
Fig. 2b shows the temporal variations in the regional average AOD values in the YRD
(covering 16 cities within the area of 25-40°N and 110-128°N). The annual mean value is
0.71±0.57. The maximum seasonal value is 0.98±0.83 in summer, followed by 0.81±0.57 in winter,
0.59±0.24 in spring, and 0.48±0.35 in autumn. Although the peak particle concentrations are
observed in winter (as shown in Fig. 3 and 5), the above results demonstrate that the maximum
regional mean AOD values occur in summer, as they reach their highest value of 1.60 in June.
This result is similar to that found by Kim et al. (2006), who reported that the value of AOD is not
only associated with the pollution levels of fine particles but is also strongly affected by other
factors (e.g., solar radiation, water vapor). The fact that the maximum AOD values occur in hot
seasons should be ascribed to the combined effects of the increase in fine aerosol production (i.e.,
due to secondary aerosol formation by gas-to-particle conversion, the hygroscopic growth of
hydrophilic aerosols or biomass burning emissions) and humid weather (Kim et al., 2006).
Consequently, the aerosol optical depth data obtained from satellite observations can reveal the
spatial distribution of aerosols to some extent, but they cannot exactly reflect pollution levels or
replace concentration data.
Figs. 2c and 2d show the spatial distributions of the annual mean particle concentrations in 16
typical cities over the YRD from December 2013 to November 2014. Generally, the spatial
distributions of $PM_{2.5}$ (Fig. 2c) and $PM_{10}$ (Fig. 2d) exhibit overall similar patterns. The annual
mean $PM_{2.5}$ and $PM_{10}$ values decrease progressively in the northwest-southeast direction, which
means that particle concentrations are comparatively high in the northwest inland areas and low in
the southeast coastal areas. The pollution levels in most cities exhibit a positive correlation with
their proximity to the sea. The farther a city is from the sea, the higher its particle concentrations
are. The maximum particle concentrations occur in Nanjing, with values of 79 $\mu g \cdot m^{-3}$ for $PM_{2.5}$
and 130 $\mu g \cdot m^{-3}$ for $PM_{10}$. Previous studies of major climatic features in the YRD have
demonstrated that the southeast coastal area is dramatically affected by the land-sea breeze and
marine air masses. The clean marine air masses are advantageous to the dilution and diffusion of
atmospheric pollutants, thus producing lighter air pollution. However, in the inland region,
clustered cities and industrial districts tend to emit more pollutants, thereby resulting in the
accumulation of more air pollutants around these cities.

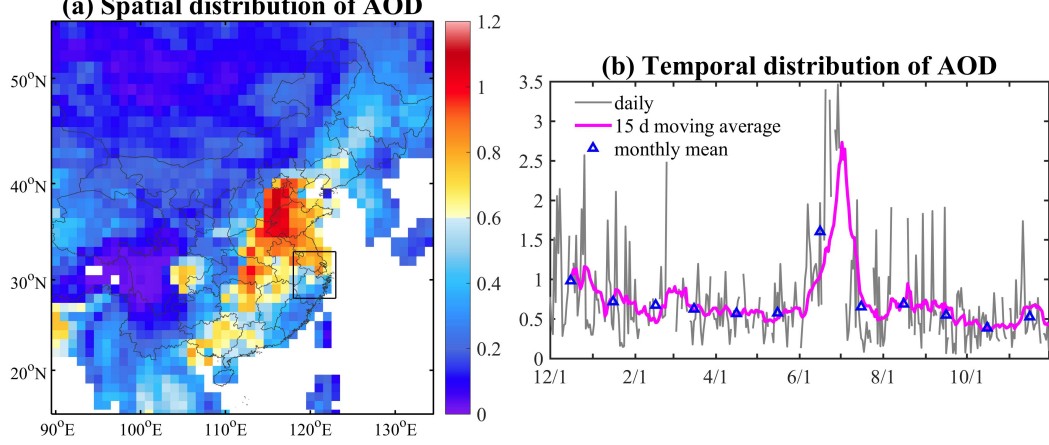


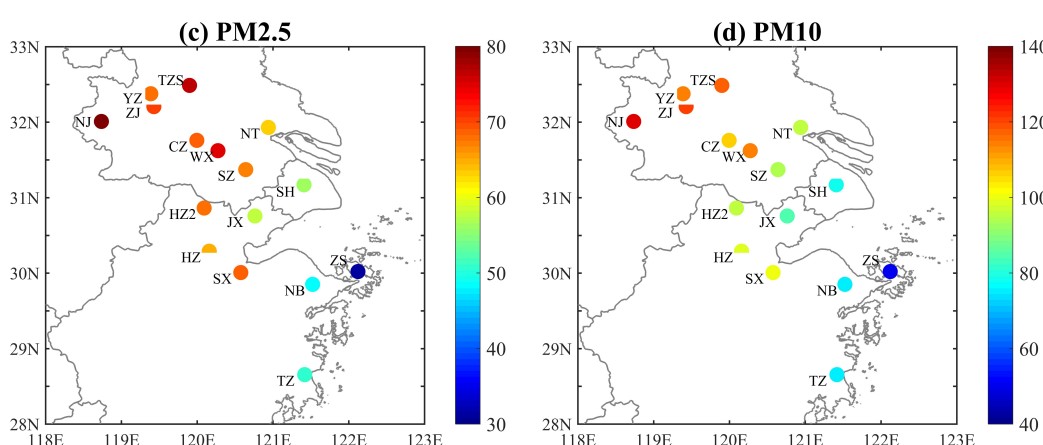



**Figure 2. The spatial distribution of annual mean AOD values (at a wavelength of 550 nm) over the**
**YRD (a); the temporal variations in regional average AOD values over 28-33ºN and 118-123ºN (b); the**
**spatial distribution of annual mean PM$_{2.5}$ concentrations (c); and the spatial distribution of annual mean**
**PM$_{10}$ concentrations (d). In (b), the gray line represents the daily value, the blue markers represent the**
**monthly mean values, and the magenta line represents the 15-day moving average value. In (c) and (d), the**
**acronyms of each city are marked, including Shanghai-SH, Changzhou-CZ, Nanjing-NJ, Nantong-NT,**
**Suzhou-SZ, Taizhoushi-TZS, Wuxi-WX, Yangzhou-YZ, Zhenjiang-ZJ, Hangzhou-HZ, Huzhou-HZ2,**
**Jiaxing-JX, Ningbo-NB, Shaoxing-SX, Taizhou-TZ, and Zhoushan-ZS.**

Fig. 3 illustrates the spatial distribution of the seasonal mean PM$_{2.5}$ in 16 cities over the YRD.
The pattern observed during each season is similar to the annual mean pattern (Fig. 2c). The PM$_{2.5}$
pollution levels are much higher in inland cities, and they decrease in the northwest-southeast
direction. PM$_{2.5}$ concentrations exhibit seasonal variations; they are highest in winter, reaching a
maximum value of 120 μg·m$^{-3}$, and they decrease throughout spring, yielding their lowest values
during summer and autumn. The difference between the PM$_{2.5}$ concentration in summer and that
in autumn is relatively small; this difference ranges from a maximum value of lower than 60
μg·m$^{-3}$ in Nanjing to a minimum value of close to 20 μg·m$^{-3}$ in Zhoushan.

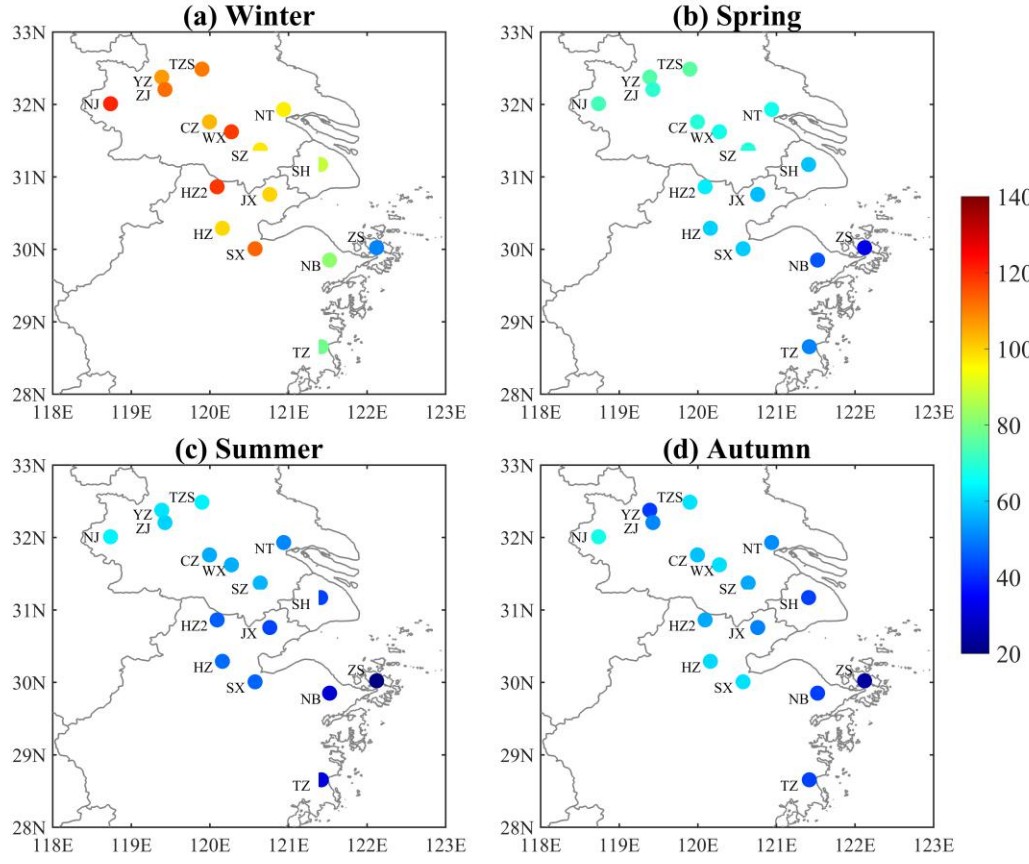

**Figure 3. The spatial distribution of seasonal mean PM$_{2.5}$ over the YRD in (a) winter, (b) spring, (c) summer,**
**and (d) autumn (unit: μg·m$^{-3}$ ). The acronyms for each city are the same as those in Figure 2.**

Table 1 quantitatively lists the annual mean concentrations of PM$_{2.5}$ and PM$_{10}$ in 16 cities
over the YRD. It also demonstrates that the particle pollution levels are relatively higher in inland
cities. The concentrations of PM$_{2.5}$ and PM$_{10}$ in 8 cities in Jiangsu Province are all higher than 60
μg·m$^{-3}$ (PM$_{2.5}$) and 80 μg·m$^{-3}$ (PM$_{10}$), respectively. However, these concentrations are
comparatively lower in the cities located in the coastal area (e.g., Ningbo, Taizhou and Zhoushan).
Only the air quality of Zhoushan meets the national standard, which may be attributed to the fact
that it is located on an island, where its air is most likely influenced by clean marine air masses.
To reveal the important role of PM$_{2.5}$ in particle pollution, the ratios of PM$_{2.5}$ concentration to
PM$_{10}$ concentration (PM$_{2.5}$/PM$_{10}$) are calculated over the YRD. As listed in Table 1, the maximum
annual mean value of the PM$_{2.5}$/PM$_{10}$ ratio is 0.72 in Shanghai, followed by Huzhou and Suzhou
(0.71), thus implying that the PM$_{2.5}$ fraction is overwhelmingly dominant relative to the PM$_{10}$
mass in these cities. The PM$_{2.5}$/PM$_{10}$ ratios in other cities range from 0.60 to 0.69, with a
minimum value of 0.58 in Zhenjiang. These values are comparable to those in other cities, such as
Beijing (He et al., 2001), Shanghai (Wang et al., 2013), Taibei (Chen et al., 1999), and Hong Kong
(Ho et al., 2003), thus suggesting that the formation of $PM_{2.5}$ from gases is the most important
source of particles in the cities of China. Table 1 also indicates that the $PM_{2.5}/PM_{10}$ ratios in all
cities exhibit distinct seasonal variation. It is remarkable that the values of $PM_{2.5}/PM_{10}$ are much
higher in winter than they are in other seasons, reaching a maximum value of 0.85 in Shanghai,
followed by a value of 0.82 in Suzhou. The highest concentrations of $PM_{2.5}$ usually occur in
winter (Fig. 3a), and high values of the $PM_{2.5}/PM_{10}$ ratio also occur during the same season (Table
1), thus indicating that $PM_{2.5}$ poses a greater threat to human health in cold seasons, which may be
related to heating activities. In summer, the values of $PM_{2.5}/PM_{10}$ in the 16 cities are medium, with
a mean value of 0.67. The lowest ratios usually occur in spring and autumn, when the mean ratios
of all cities are 0.61 (spring) and 0.63 (autumn). The minimum value occurs in the autumn in
Yangzhou, with a value of 0.51, followed by a value of 0.52 in the spring in Nanjing and the
autumn in Zhenjiang. The above discussion of the spatial and temporal variations in $PM_{2.5}/PM_{10}$
ratios also implies that particles originate from various kinds of sources and are variedly emitted.

**Table 1. Annual mean concentrations of $PM_{2.5}$ and $PM_{10}$, and the annual and seasonal mean values of $PM_{2.5}/$**
**$PM_{10}$ ratio, in 16 cities over the YRD.**

| Cities | | $PM_{2.5}$ ($\mu g \cdot m^{-3}$) | $PM_{10}$ ($\mu g \cdot m^{-3}$) | $PM_{2.5}/PM_{10}$ | | | | |
| --- | --- | --- | --- | --- | --- | --- | --- | --- |
| | | | | Annual | Winter | Spring | Summer | Autumn |
| Shanghai | | 56 | 78 | 0.72 | 0.85 | 0.68 | 0.72 | 0.66 |
| Jiangsu Province | Nanjing | 79 | 130 | 0.61 | 0.64 | 0.52 | 0.70 | 0.60 |
| | Changzhou | 69 | 106 | 0.65 | 0.73 | 0.60 | 0.67 | 0.62 |
| | Nantong | 63 | 95 | 0.66 | 0.72 | 0.62 | 0.71 | 0.64 |
| | Suzhou | 67 | 94 | 0.71 | 0.82 | 0.68 | 0.71 | 0.67 |
| | Taizhoushi | 76 | 117 | 0.65 | 0.66 | 0.58 | 0.72 | 0.66 |
| | Wuxi | 75 | 114 | 0.66 | 0.73 | 0.59 | 0.67 | 0.62 |
| | Yangzhou | 68 | 114 | 0.60 | 0.69 | 0.58 | 0.59 | 0.51 |
| | Zhenjiang | 70 | 121 | 0.58 | 0.71 | 0.54 | 0.58 | 0.52 |
| Zhejiang Province | Hangzhou | 65 | 99 | 0.66 | 0.74 | 0.59 | 0.63 | 0.66 |
| | Huzhou | 68 | 96 | 0.71 | 0.78 | 0.66 | 0.68 | 0.69 |
| | Jiaxing | 58 | 84 | 0.69 | 0.75 | 0.65 | 0.68 | 0.69 |
| | Ningbo | 48 | 75 | 0.64 | 0.69 | 0.62 | 0.63 | 0.62 |
| | Shaoxing | 68 | 100 | 0.68 | 0.72 | 0.62 | 0.71 | 0.68 |

| | | | | | | |
|---|---|---|---|---|---|---|
| Taizhou | 50 | 75 | 0.67 | 0.69 | 0.66 | 0.66 | 0.65 |
| Zhoushan | 31 | 50 | 0.63 | 0.66 | 0.62 | 0.66 | 0.55 |

### 3.1.2 Temporal variations in particle pollution

Fig. 4 shows the annual mean diurnal variations in $PM_{2.5}$ (Fig. 4a) and $PM_{10}$ (Fig. 4b) in 16 cities over the YRD. Obviously, the diurnal cycles of particle concentrations in most cities follow a similar pattern. The $PM_{2.5}$ concentrations maintain comparably high values from 0:00 to 8:00. Then, coinciding with more vehicle emissions during rush hours, these concentrations increase rapidly from 8:00 to 12:00. After reaching their peak, the $PM_{2.5}$ concentrations decrease and remain at low values until sunset. During nighttime, the pollutants accumulate until midnight, which can be attributed to the more stable atmospheric stratification in the boundary layer. In comparison, there are two peaks in the diurnal cycles of the $PM_{10}$ concentrations in several cities. The broad morning peak of $PM_{10}$ concentrations is more evident from 8:00 to 12:00, and the evening peak occurs at approximately 20:00. In addition, the diurnal change in particle concentrations in the southeast coastal area, such as Zhoushan, is much smaller. As discussed in Section 3.1.1, this difference might be related to its special geographic location, which exhibits fewer emissions of precursors and lower pollution levels.

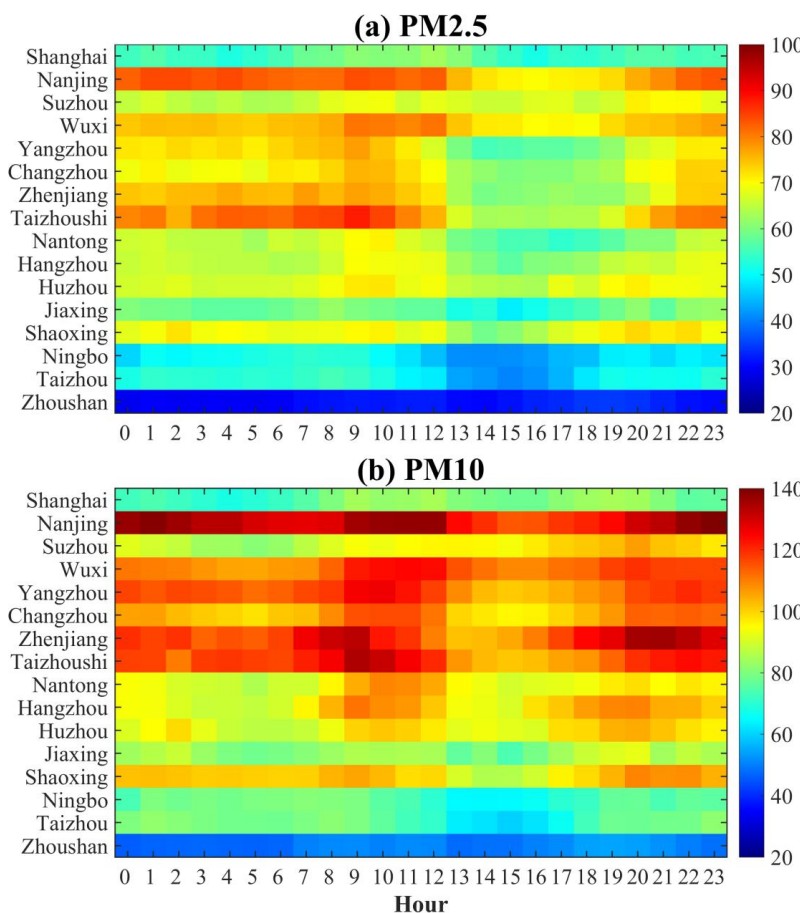

**(a) PM2.5**

**(b) PM10**

Hour

Figure 4. Diurnal variations in PM$_{2.5}$ (a) and PM$_{10}$ (b) concentrations in 16 cities of the YRD (unit: μg·m$^{-3}$).

Fig. 5 shows the monthly mean concentrations of PM$_{2.5}$ and PM$_{10}$ in 16 cities of the YRD. As illustrated in this figure, there are three peaks in the seasonal variations in particles. These three peaks occur in December, March, and May/June. This monthly variation pattern is more obvious for PM$_{10}$. The causes resulting in the wintertime peak of particle concentrations can be explained by two factors. One is the enhanced emissions of pollutants from residential heating. The other is the stable and poor meteorological conditions that limit the diffusion of atmospheric pollutants. The drivers of the peak appearing in March may be associated with dust storm events in spring (Zhuang et al., 2001; Fu et al., 2010; 2014). As discussed in Section 3.1.1, the values of the PM$_{2.5}$/PM$_{10}$ ratio in 16 cities are lowest in spring, with a mean ratio of 0.61. High PM$_{10}$ concentrations during this period further demonstrate that dust storms can bring more coarse dust particles to the YRD. The peak in May/June is probably caused by the field burning of crop residue in rural areas of China, which is regarded to be an important source of biomass burning (Yan et al., 2006; Yang et al., 2007; Zhu et al., 2012).

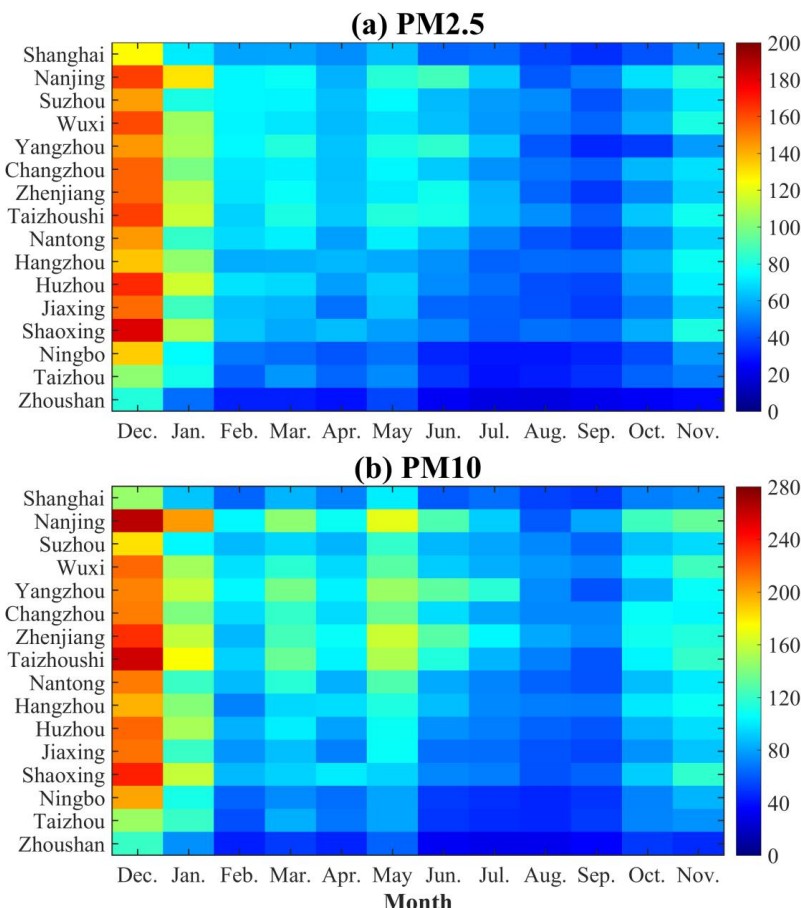

**(a) PM2.5**

**(b) PM10**

**Figure 5. Monthly variations in PM$_{2.5}$ (a) and PM$_{10}$ (b) concentrations in 16 cities of the YRD (unit: μg·m$^{-3}$).**

**3.1.3 Regional severe particle pollution in the YRD**
According to the National Ambient Air Quality Standard (NAAQS) of China, urban air
quality must meet the second standard, with daily mean concentrations of PM$_{2.5}$ and PM$_{10}$ that are
lower than 75 μg·m$^{-3}$ and 150 μg·m$^{-3}$, respectively. In this study, when the daily mean PM$_{2.5}$ (PM$_{10}$)
concentrations exceed the national air quality standard in most (i.e., 8 or more) of the 16 cities, we
define this as large-scale regional PM$_{2.5}$ (PM$_{10}$) pollution. Consequently, from December 2013 to
November 2014, there were 98 (46) days when large-scale regional PM$_{2.5}$ (PM$_{10}$) pollution
episodes were identified. That is, the YRD suffered from regional PM$_{2.5}$ (PM$_{10}$) pollution during
nearly 28.0% (13.1%) of the days of the year.
Table 2 shows the typical regional severe particle pollution episodes (that lasted no less than
3 days) in the YRD from December 2013 to November 2014. As illustrated in this table, dozens of
continuous large-scale particle pollution episodes occurred. For example, PM$_{2.5}$ concentrations
exceeded the national standard in all 16 cities from December 1 to 5, 2013, and there were more
than 14 cities facing heavy PM$_{10}$ pollution at the same time. From May 26 to 30, 2014, serious

PM$_{2.5}$ and PM$_{10}$ pollution episodes were observed in more than 10 cities. It appears that high-PM$_{2.5}$ pollution episodes are remarkably associated with high-PM$_{10}$ pollution episodes. Moreover, regional PM$_{2.5}$ pollution episodes occurred much more frequently than PM$_{10}$ pollution episodes. This may be due to the fact that fine particles dominate the composition of particles in the YRD (as discussed in Section 3.1.2).

**Table 2. The typical regional severe particle pollution episodes (lasting for no less than 3 days) in the YRD from December 2013 to November 2014.**

| Episodes of PM$_{2.5}$ pollution | Episodes of PM$_{10}$ pollution |
|---|---|
| 1-6 Dec. | 1-6 Dec. |
| 11-15 Dec. | 12-15 Dec. |
| 24-26 Dec. | 24-26 Dec. |
| 28 Dec. - 6 Jan. | 29 Dec. - 5 Jan. |
| 15-20 Jan. | 17-20 Dec. |
| 30 Jan. - 2 Feb. | 26-30 May |
| 20-24 Feb. | |
| 16-18 Mar. | |
| 8-10 Apr. | |
| 20-22 May | |
| 26-30 May | |
| 5-7 Jun. | |
| 28 Jun. - 1 Jul. | |
| 10-12 Nov. | |

**3.2 Synoptic weather classification**

In this study, to examine the relationship between regional severe particle pollution in the YRD and weather situations, synoptic weather classification is carried out from December 2013 to November 2014. Using the method described in Section 2.2, we conduct the classification of the synoptic weather pattern by using the dataset of geopotential height at 850 hPa collected from the NCEP reanalysis data. As shown in Table 3, five weather patterns are finally identified. Unknown patterns are defined as 'the unclassified pattern'. The weather situation on 95.6% of the days during the study period is classified as one of the five typical synoptic weather patterns.

Table 3 lists the typical date, number of days, and seasonal occurrence frequencies of each synoptic weather pattern. As demonstrated in this table, Pattern 1 is the dominant weather pattern in the YRD, which accounts for 47.6% of all of the days of the year (from December 2013 to

November 2014). The occurrence frequencies of Patterns 2 and 3 are 20.0% and 18.1%,
respectively. Patterns 4 and 5 are identified on the fewest number of days, with occurrence
frequencies of 4.1% and 5.8%, respectively.

Table 3 also shows the seasonal occurrence frequencies of each pattern from December 2013

to November 2014. Obviously, they are distinctly different. Pattern 1 tends to occur in winter, with
a frequency of 30.5%, followed by spring (25.9%), summer (21.8%) and autumn (21.8%). Pattern
2 is the most popular weather pattern in summer, with an occurrence frequency of 37.0%,
followed by spring (30.1%), autumn (21.9%) and winter (11.0%). For Pattern 3, the seasonal
frequencies occur in the order of winter (36.4%), spring (27.3%), autumn (19.7%) and summer
(16.7%). Both Pattern 4 and Pattern 5 are most likely to occur in autumn, with occurrence
frequencies of 53.3% and 42.9%, respectively. The occurrence frequencies of Pattern 4 and Pattern
5 during other seasons account for nearly 50%.

**Table 3. The typical date, number of days, and seasonal occurrence frequencies of each synoptic weather**
**pattern.**

| Type | Typical date | Number of days | Occurrence frequency (%) | | | |
|---|---|---|---|---|---|---|
| | | | Spring | Summer | Autumn | Winter |
| Pattern 1 | 2014-05-12 | 174 (47.7%) | 25.9 | 21.8 | 21.8 | 30.5 |
| Pattern 2 | 2014-05-09 | 73 (20.0%) | 30.1 | 37.0 | 21.9 | 11.0 |
| Pattern 3 | 2014-02-18 | 66 (18.1%) | 27.3 | 16.7 | 19.7 | 36.4 |
| Pattern 4 | 2014-10-07 | 15 (4.1%) | 13.3 | 26.7 | 53.3 | 6.7 |
| Pattern 5 | 2014-09-14 | 21 (5.8%) | 19.0 | 23.8 | 42.9 | 14.3 |
| Unclassified pattern | — | 16 (4.4%) | — | — | — | — |


**3.3 Effects of synoptic weather patterns on particle pollution**
**3.3.1 Relationship between synoptic weather pattern and particle pollution**

To determine the relationship between synoptic weather patterns and particle pollution, the

occurrence frequencies of the five typical synoptic patterns during the regional severe particle
pollution episodes are calculated. As shown in Table 4, during the days with regional $PM_{2.5}$ ($PM_{10}$)
pollution episodes, Pattern 1 is the dominant synoptic weather pattern, with an occurrence
frequency of 70.4% (78.3%). Pattern 2 and Pattern 3 both occur on 14.3% of the days with $PM_{2.5}$
pollution episodes. During $PM_{10}$ pollution episodes, Pattern 2 (6.5%) appears less frequently than
Pattern 3 (15.2%). The occurrence frequencies of Pattern 4 and Pattern 5 are less than 1% and can
thus almost be ignored.

According to Table 3 and Table 4, the occurrence frequency of Pattern 1 during regional

particle pollution episodes is obviously higher than its occurrence during the entire year. In
contrast, the occurrences of Pattern 2 and Pattern 3 during these episodes are less frequent than
those throughout the year. Moreover, Pattern 4 and Pattern 5 appear far less frequently during
regional particle pollution episodes than they do throughout the year. In summary, these data
suggest that the weather situation of Pattern 1 is more beneficial for the formation of large-scale
regional particle pollution in the YRD.

**Table 4. The occurrence frequencies of synoptic weather patterns during regional severe $PM_{2.5}$ and $PM_{10}$**
**pollution episodes**

| Type | $PM_{2.5}$ | | $PM_{10}$ | |
|---|---|---|---|---|
| | Number of days | Frequency (%) | Number of days | Frequency (%) |
| Pattern 1 | 69 | 70.4 | 36 | 78.3 |
| Pattern 2 | 14 | 14.3 | 3 | 6.5 |
| Pattern 3 | 14 | 14.3 | 7 | 15.2 |
| Pattern 4 | 0 | 0% | 0 | 0 |
| Pattern 5 | 1 | 1.0 | 0 | 0 |


Fig. 6 shows the box-and-whisker plot of the mean concentrations of air pollutants ($PM_{10}$,

$PM_{2.5}$, $O_3$, $NO_2$, $SO_2$ and CO) and the meteorological parameters (WS, T, P and RH) of 16 cities
under the five synoptic weather patterns, as well as the corresponding spatial distribution of AOD
over eastern China. These statistical results are also listed in Table 5.

As shown in Figs. 6a-6f and Table 5, the highest average concentrations of the main air

pollutants (except for $O_3$) in the 16 cities in the YRD are associated with Pattern 1. Since aerosols
can reflect and absorb solar radiation and thereby cause the photochemical production of $O_3$ to
decrease (Kaufman et al, 2002), the $O_3$ concentration is lowest for Pattern 1 (Fig. 6c). As
mentioned above, Pattern 1 is most likely to occur during winter (30.5%) and spring (25.9%).
Therefore, the weather situation of this pattern features the weakest surface wind, lowest humidity,
second-highest surface pressure, and low temperature. All of these weather characteristics are
conducive to the accumulation of particles and their precursors (i.e., $SO_2$, $NO_2$ and CO). For
Pattern 3, the concentrations of $PM_{10}$, $PM_{2.5}$ $NO_2$ and $SO_2$ are the second-highest compared to
those of the other patterns. This pattern features the highest surface pressure and much stronger
surface wind. The temperature is the lowest, as Pattern 3 also tends to occur during winter (37.0%)
and spring (30.1%). Under the weather situation of Pattern 1 and Pattern 3, the YRD is both under
the control of high pressure and likely to suffer serious particle pollution. The strength of the
surface wind for different weather patterns plays a key role in the occurrence frequency of
regional severe particle pollution episodes. Pattern 1, which has the weakest surface wind, is
regarded as 'the most polluted pattern'. The pollution levels of the main pollutants in Pattern 2 are
in the middle and slightly lower than those of Pattern 3. Due to its high occurrence frequency in
summer (37.0%) and spring (30.1), the weather condition of Pattern 2 is characterized by its
relatively high temperature, low pressure, and the lowest RH. In contrast, Pattern 4 and Pattern 5
are 'the clean patterns', in which the concentrations of all of their pollutants are distinctly lower
than those of the other three patterns. Their meteorological conditions of relatively high humidity,
high temperature, strong wind (especially for Pattern 5) and much lower surface pressure are also
favorable for the mitigation of pollutants.
Figs. 6k to 6o display the spatial distribution of AOD over eastern China under different
synoptic weather patterns. The regional mean values of AOD in the YRD (28-33°N, 118-123°N)
are 0.74 for Pattern 1, 0.64 for Pattern 2, 0.81 for Pattern 3, 0.47 for Pattern 4 and 0.49 for Pattern
5. Additionally, AOD is higher over the YRD for Pattern 3, Pattern 1 and Pattern 2. For these three
patterns, high AOD values usually occur in the BTH, the YRD, and the SCB, as well as the
provinces of Shanxi, Shandong, Hubei, Hunan, Anhui and Guangxi. The highest AOD values are
mainly found in northeastern China. For Pattern 4 and Pattern 5, high AOD values are mostly
concentrated in the BTH and Shandong Province, while relatively low AOD values are found in
the YRD. Since AOD is closely related to the concentrations of fine particles, it can be concluded
that the YRD is most heavily polluted under the weather situations of Pattern 1 and Pattern 3.

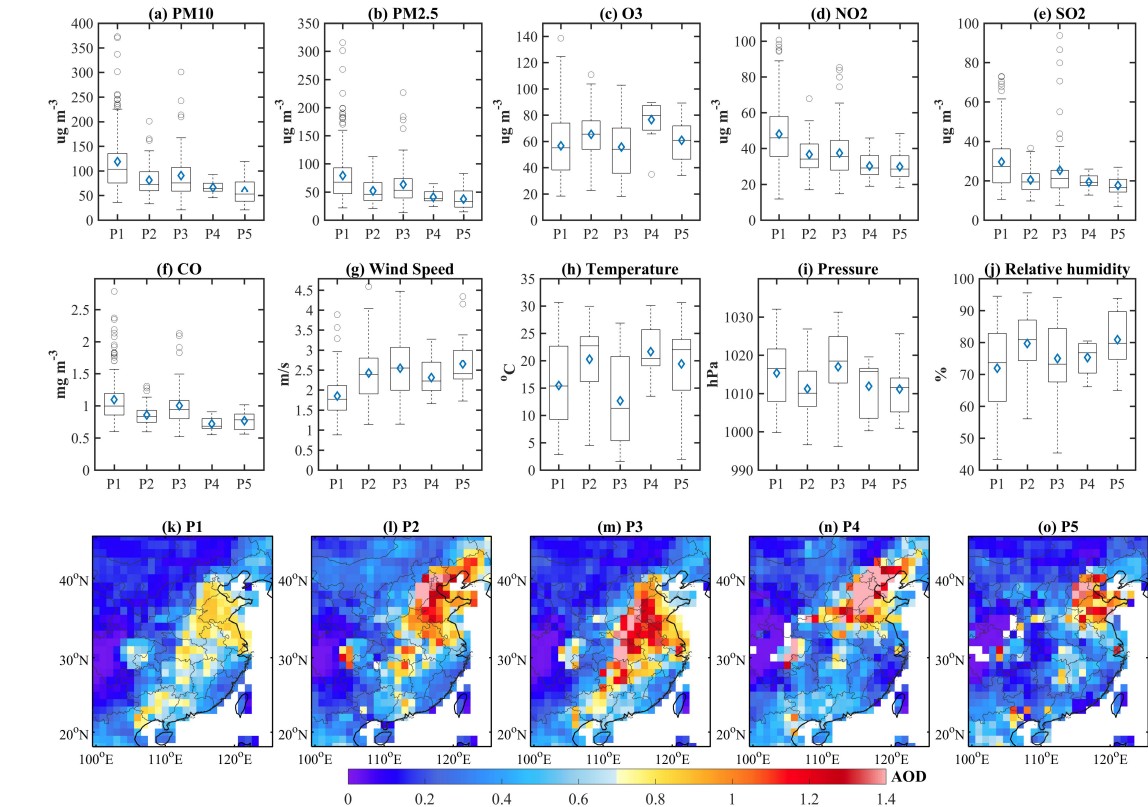

**Figure 6.** (a-j) Box-and-whisker plots for the mean values of air pollutant concentrations and meteorological parameters of 16 typical YRD cities. The edges of each box in (a-j) are the 25[th] and 75[th] percentiles; the band inside the box is the median; the diamond is the average; and the whiskers extend to the most extreme data values. (k-p) Spatial distributions of AOD for the five synoptic weather patterns. P1, P2, P3, P4, and P5 represent Pattern 1, Pattern 2, Pattern 3, Pattern 4, and Pattern 5, respectively.

**Table 5.** The average values of air pollutant concentrations and meteorological factors for the 16 typical YRD cities under different synoptic weather patterns.

| Type | $PM_{10}$ | $PM_{2.5}$ | $O_3$ | $NO_2$ | $SO_2$ | CO | $SO_2$ | WS | T | P | RH |
|---|---|---|---|---|---|---|---|---|---|---|---|
| Pattern 1 | 116.5±66.9 | 75.9±49.9 | 57.7±27.3 | 46.9±19.2 | 29.3±17.1 | 1.08±0.41 | 29.3±17.1 | 1.84±0.67 | 15.8±7.8 | 1015.0±8.5 | 72.3±14.4 |
| Pattern 2 | 81.5±38.4 | 52.3±27.4 | 65.5±23.6 | 36.1±13.4 | 20.6±9.9 | 0.86±0.24 | 20.6±9.9 | 2.38±0.88 | 20.3±6.3 | 1011.2±6.7 | 79.8±10.2 |
| Pattern 3 | 86.9±49.5 | 59.1±37.3 | 58.5±25.5 | 35.1±15.5 | 23.3±15.9 | 0.96±0.35 | 23.3±15.9 | 2.59±0.87 | 13.4±8.2 | 1016.1±9.6 | 76.0±11.6 |
| Pattern 4 | 66.1±18.8 | 40.7±15.9 | 76.8±19.6 | 29.4±9.8 | 19.4±6.4 | 0.72±0.17 | 19.4±6.4 | 2.29±0.64 | 21.7±4.9 | 1011.8±7.0 | 75.4±5.8 |
| Pattern 5 | 58.7±31.3 | 37.4±22.5 | 61.1±20.6 | 29.1±11.1 | 17.8±8.4 | 0.77±0.22 | 17.8±8.4 | 2.63±0.93 | 19.4±8.0 | 1011.1±6.9 | 81.0±9.8 |

### 3.3.2 The impact mechanism of synoptic weather patterns on severe particle pollution

Figs. 7-11 present the meteorological fields and backward trajectories under the weather situations of the Pattern 1 (northwestly inland wind), Pattern 2 (southwestly), Pattern 3 (northly inland wind), Pattern 4 (cyclone-related) and Pattern 5 (oceanic circulation related). The first two graphs of Figs. 7-11 illustrate the 850 hPa and 500 hPa geopotential height field and wind field, respectively. The third graphs display the sea level pressure field and 1000 hPa wind field. The

highlighted boxes denote the study area (i.e., the YRD). The fourth graphs demonstrate the
height-latitude cross-sections of vertical velocity over the latitudes of 25-40$^o$N, which are
averaged from the longitudes of 110-128$^o$E. The bold black lines show the latitude range of 16
cities (28.6-32.5$^o$N) over the YRD. The positive wind speeds ($10^2$ Pa s$^{-1}$) represent vertical
downward atmospheric motions, while the negative wind speeds represent upward motions. In
addition, it is well known that atmospheric pollutant transport trajectories are deeply affected by
synoptic systems. As shown in the fifth graphs in Figs. 7-11, to reveal how the typical synoptic
weather patterns influence the distribution of particles in the YRD, the 72-h backward trajectories
are calculated and then clustered. Given that Nanjing is the most polluted city in the YRD, as
described in Section 3.1, the observational site in Nanjing (32$^o$N, 118.8$^o$E) is chosen for the
terminus of the trajectory of each synoptic weather pattern.
As illustrated in Fig. 7a, Pattern 1 usually occurs when the YRD is located at the rear of the
East Asian major trough and is under the control of a high-pressure ridge at 850 hPa. The center of
the high-pressure system is located in the northwestern Pacific Ocean. Meanwhile, northeastern
China is strongly affected by a low-pressure system, namely, the Aleutian Low. The strong
horizontal northwest wind at the rear of the East Asian major trough can transport pollutants from
the BTH (with high AOD, as shown in Fig. 6k) to the YRD. At the same time, the west and
southwest wind at the rear of the high-pressure ridge can also transport pollutants from central and
southwestern China (such as the SCB and Guangxi Province) to the YRD. The confluence of air
flows may cause an accumulation of pollutants in the YRD. Accordingly, the atmospheric
circulation at 500 hPa features a shallow through with a west-northwest flow (Fig. 7b). The sea
level pressure pattern is nearly dominated by a uniform pressure field, which exhibits relatively
weak anti-cyclonic circulation over the YRD (Fig. 7c). The above discussion can be further
explained by the 72-h backward trajectories displayed in Fig. 7e. When the YRD is under the
control of Pattern 1, the air masses are mainly from northern China (44%), followed by the central
(36%) and northeastern regions of the YRD (19%). This suggests that particle pollution is
remarkably affected by the polluted air masses from the BTH and the central city clusters. Surface
meteorological observation records also indicate that west-northwest-southwest surface winds are
dominant in Nanjing (Fig. 7f) and that high PM$_{2.5}$ is closely associated with the transport of
polluted air masses in these wind directions. In the vertical section (Fig. 7d), the relatively weak
upward air flows are dominant to the south of 30°N, while clear downward air flows are prevalent
to the north of 30°N. The largest descending velocity (~8×10⁻² Pa s⁻¹) appears at an altitude of 500
hPa and a latitude of 37.5°N. Downward motion is dominant above the YRD, which is in
accordance with the 850 hPa circulation pattern represented by a high-pressure ridge. Thus, the
weather conditions are relatively stable near the surface, which is beneficial to the local
accumulation of pollutants. Overall, Pattern 1 represents a stable synoptic weather pattern that is
extremely conducive to the build-up of atmospheric pollutants over the YRD. This result is
consistent with the findings of Zheng et al (2015b).

# Pattern 1

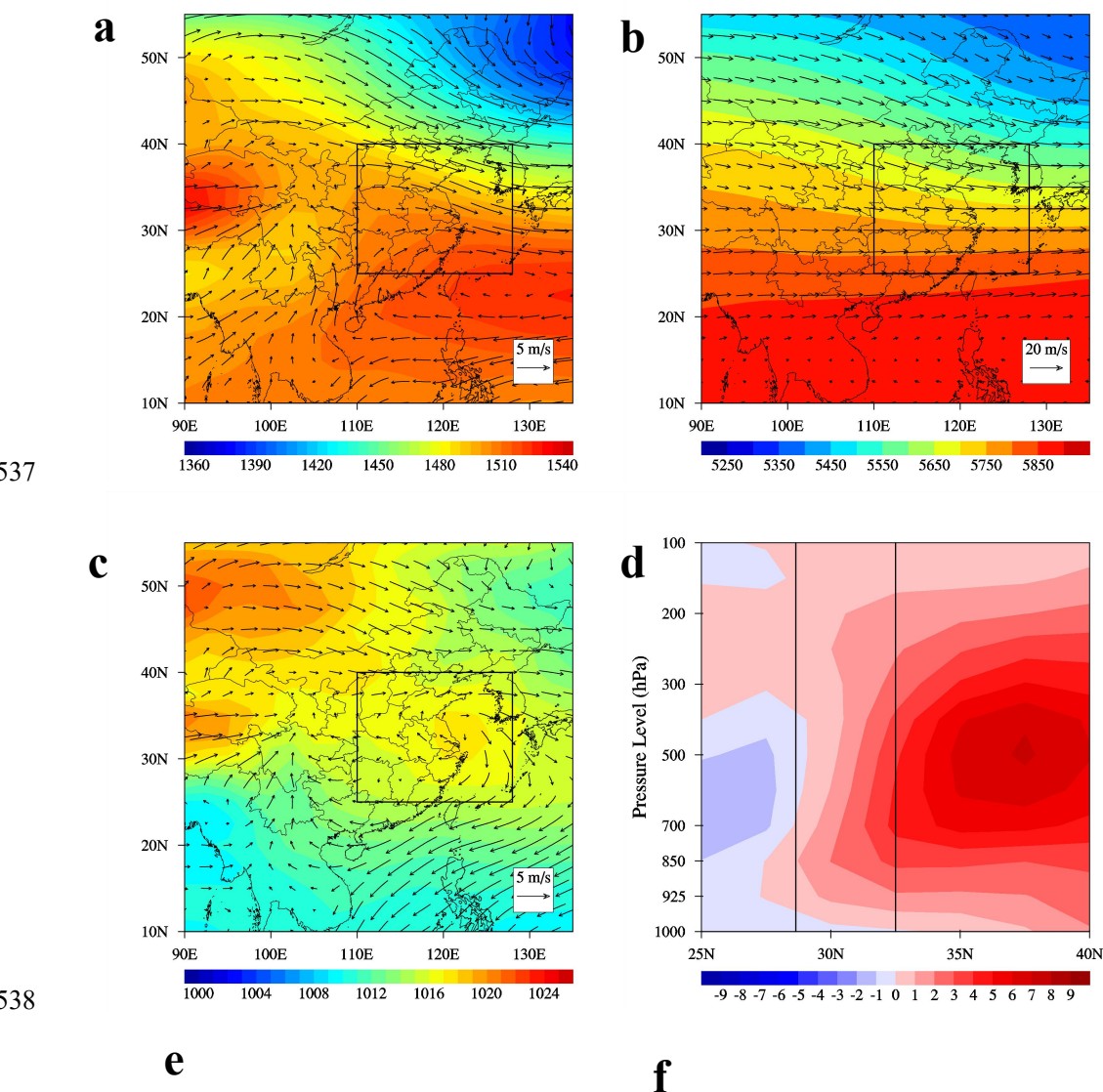



e                                            f

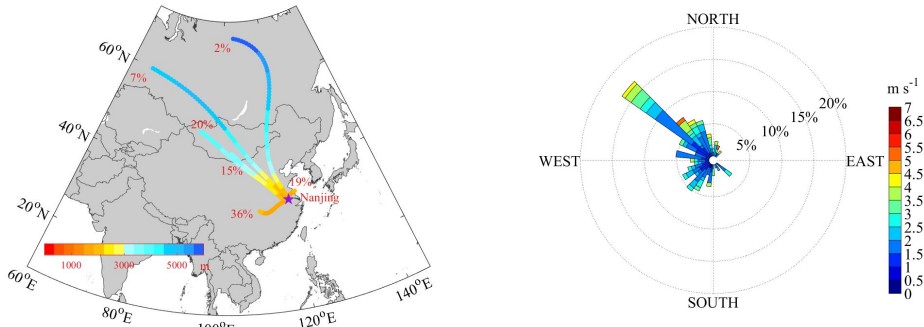



**Figure 7. Weather condition in Pattern 1. (a) 850 hPa geopotential height field and wind field; (b) 500 hPa**
**geopotential height field and wind field; (c) sea level pressure field and 1000 hPa wind field; (d)**
**height-latitude cross-sections of vertical velocity (unit: $10^{-2}$ Pa/s) averaged from longitude of 110-128ºE; (e)**
**72-h backward trajectory ending at a height of 1500 m; and (f) observation wind rose plots in Nanjing. In**
**(a)-(c), the highlighted boxes denote the study area (i.e., the YRD). In (d), the black rectangular region**
**represents the 16 cities in the YRD (28.6-32.5ºN). In (e), the purple marker indicates the location of Nanjing**
**(32ºN, 118.8ºE). These data represent averages for all days corresponding to Pattern 1.**

In Pattern 2, a low-pressure center (the Southeast Vortex) is centered in the SCB, the East
China Sea is influenced by a high-pressure system, and a depression inverted trough extends and
covers the YRD region at a latitude at 850 hPa (Fig. 8a). Consequently, in the YRD, the strong
southwest air flows from southern China meet with the southeast air flows from the East China
Sea. After the convergence of these air masses, they jointly transport pollutants northwestward. In
contrast, at the surface (Fig. 8c), the study area is located at the bottom of a high-pressure system
and is impacted by a strong southeast wind. In the middle troposphere (Fig. 8b), the sparse
isopleths indicate that there is a small geopotential height gradient, while the shallow ridge causes
westerly flows. Fig. 8e also illustrates these air pollutant transport paths. For the days when
Pattern 2 is dominant, approximately 42% of the air masses are from the southwest and the south
of China, and 15% are from the East China Sea. The air masses from the East China Sea are very
important because the clean marine air masses may dilute the particle concentrations in the YRD.
In addition, nearly 43% of air masses originate from the local sources of the YRD, which may be
related to their short-range transport in the northwest direction. This is also in accordance with the
dominant northwest surface wind in Nanjing (Fig. 8f). In regard to its vertical structure (Fig. 8d),
Pattern 2 is obviously different than Pattern 1, as upward air flows are dominant to the south of
37.5ºN. The largest updraft zone ($\sim 7 \times 10^{-2}$ Pa s$^{-1}$) appears above the YRD and between the

altitudes of 700 hPa and 500 hPa. The vertical velocity close to the surface is weaker than that at higher levels over the YRD. Meanwhile, stronger upward motion occurs near the surface at a latitude of 37.5°N, with weak downward motion occurring above the 700 hPa layer. The above discussion suggests that atmospheric pollutants in the YRD are horizontally transported northwestward to a higher latitude and vertically transported upward to higher layers. Therefore, despite the transport of abundant pollutants to the YRD via southwest air flows and the short-range northwest transport of polluted air masses, the strong surface southeast wind and upward motion under the weather situation of Pattern 2 result in much less particle pollution over the YRD compared to Pattern 1.

# Pattern 2

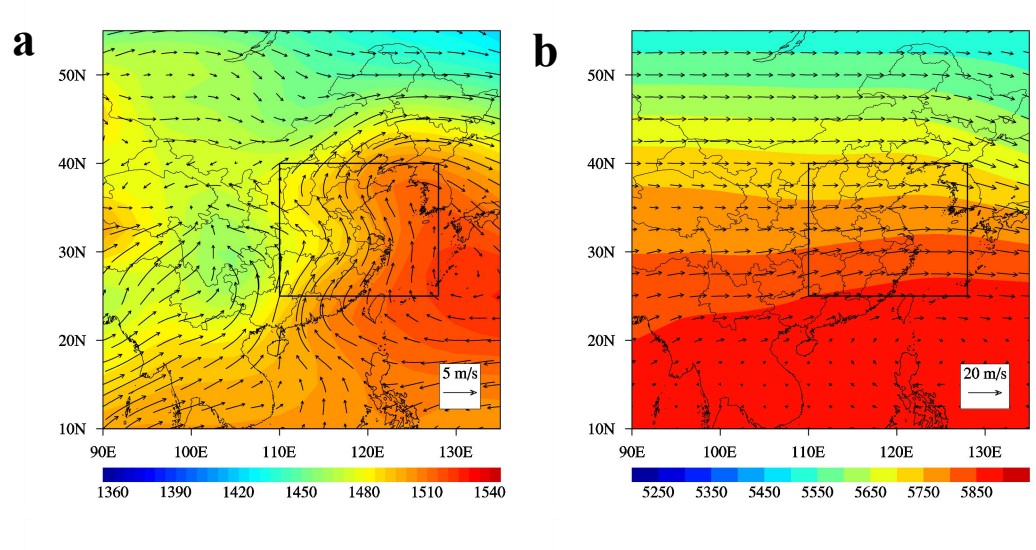

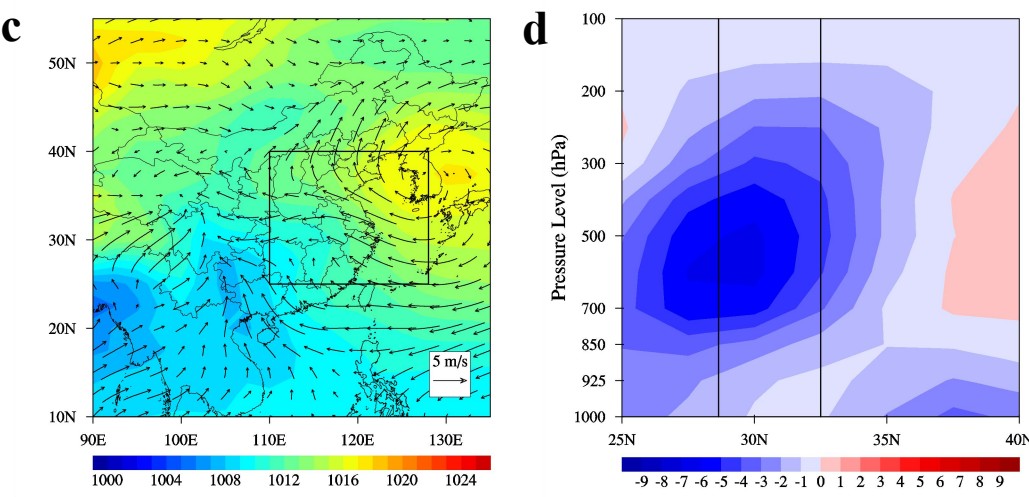

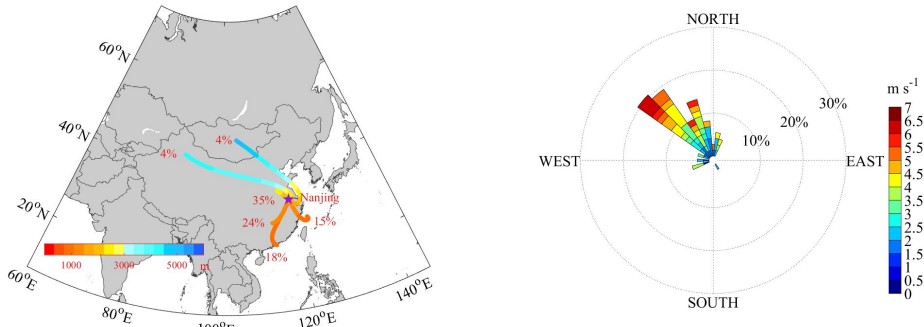


**Figure 8. As in Fig. 7, but for Pattern 2.**

Pattern 3 tends to occur in winter (36.4%, as displayed in Table 3). Under this circumstance,

the YRD is mainly controlled by a high-pressure system that is centered in central China (Fig. 9a).
Meanwhile, northeastern China is under the steering influence of the northwest air flows at the
rear of the East Asian major trough, with its trough axis appearing along the eastern coastline of
China. Affected by the strong northwest winds coming from northern China, the polluted air
masses from the BTH are easily transported to the YRD. At the higher layer of 500 hPa (Fig. 9b),
the circulation structure patterns are similar to those of Pattern 1. A trough appears in the upper
atmosphere, resulting in relatively strong west-northwest flows. The presence of dense isopleths
indicates that there is a large geopotential height gradient and strong downward flows. At the
surface layer (Fig. 9c), the presence of strong northerly wind is also evident, and the YRD is
located at the bottom of a high-pressure system centered in the remote Mongolian region. The
above discussion is further supported by the results of back trajectory calculations. As suggested
in Fig. 9e, most air masses in clusters are from the Loess Plateau (31%). The transport path of this
cluster is relatively short, which may be attributed to its strong anti-cyclonic circulation. Due to
the strong northerly wind, the long-range transport of air masses from remote Mongolia and
northern China account for 22% and 18% of all trajectories, respectively. In addition, the local
transport of air masses from the southeast coastal area in the YRD accounts for 26% of all
trajectories, and the marine air masses cluster that originates from the western Pacific via the
Yellow Sea accounts for 4% of all trajectories. For the vertical structure (Fig. 9d), the distribution
of the vertical flow field is similar to that of Pattern 1, whereas the vertical wind is slightly
stronger in the weather system of Pattern 3. Due to the influence of the high-pressure system,
downward air flows are dominant to the north of approximately 28°N (including the YRD) below

an altitude of 300 hPa. The largest descending velocity (~$9 \times 10^{-2}$ Pa s$^{-1}$) also appears at an altitude of 500 hPa, covering the latitude of 35-40$^{\circ}$N. However, despite the higher surface pressure (Figs. 6i and 9c) and stronger downward motion (Fig. 9d), the surface wind is also much stronger for Pattern 3 (Figs. 6g, 9a and 9c), which alleviates the problems of air pollution over the YRD compared to Pattern 1. Overall, under the weather situation of Pattern 3, the strong northwest wind in the front of the high-pressure system usually leads to the transport of polluted air masses from the BTH to the YRD. Nevertheless, the strong surface wind is conducive to the mitigation of pollutants, which plays a significant role in the level of air pollution over the YRD.

## Pattern 3

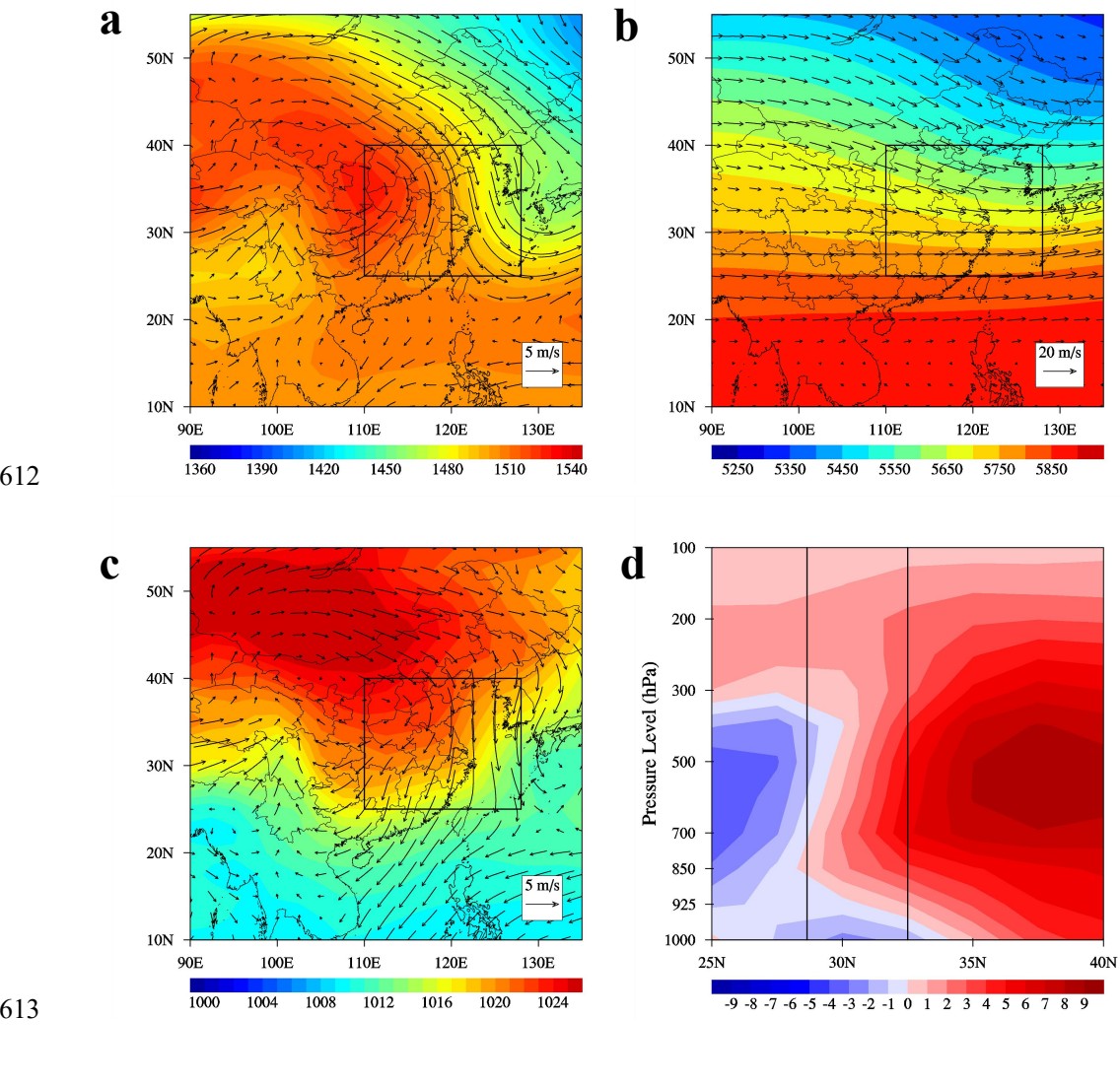

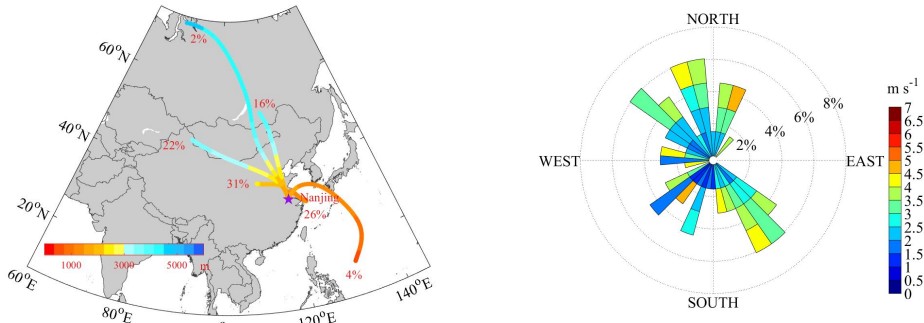


**Figure 9. As in Fig. 7, but for Pattern 3.**

In Pattern 4, on both the surface and at the 850 hPa level, the study area is under the control
of a high-pressure system (Figs. 10a and 10c). The center of the high-pressure system is located in
the Sea of Japan, while a cyclonic circulation occurs over the Philippine Sea. Anti-cyclonic
circulation prevails over the YRD and horizontally brings the clean marine air masses to the land.
Meanwhile, the sparse isopleths represent a small geopotential height gradient in the middle
troposphere, which is accompanied by a much weaker west wind compared to the other patterns
(Fig. 10b). Accordingly, influenced by the high-pressure system, downward atmospheric motion is
clearly dominant in the vertical direction (Fig. 10d). The strongest downward motion ($\sim 6 \times 10^{-2}$ Pa
$s^{-1}$) appears between the altitudes of 300 hPa and 500 hPa at a latitude of $35^{\circ}$N. The weak updrafts
near the surface may be related to the regional thermodynamic circulation. As shown in Fig. 10e,
the cluster with the largest frequency of 32% represents the local transport of air masses from the
southern adjacent areas in the YRD. Additionally, the air masses originating from northern China
via the Bohai Bay (25%), from Japan via the Yellow Sea (23%), and from the Philippines via the
East China Sea (5%) are also representative. In total, the clusters that pass over the ocean areas
account for more than 50% of all trajectories. Therefore, under this weather situation, the dilution
effects of clean marine air masses play a large role in the particle pollution over the YRD.
Pattern 5 features one of the most complex circulation situations at 850 hPa (Fig. 11a). The
YRD is located between the bottom of the northern high-pressure system and the top of the
southern weak low-pressure system. Thus, the strong horizontal east wind prevails and easily
carries clean marine air masses from the East China Sea to the YRD. The corresponding
circulation structure at the surface layer is similar to that at the 850 hPa layer (Fig. 11c), while

east-northeast flows are prevalent over the study domain. In the upper troposphere, a ridge appears in the east due to the tropical cyclonic system, thus leading to the west-southwest flows over the region. Due to the abovementioned two opposite pressure systems (Fig. 11a), strong upward air flows are dominant to the south of the latitude of 35 ᵒN, while downward motion is obvious in the north (Fig. 11d). The largest ascending velocity ($\sim -9\times10^{-2}$ Pa s$^{-1}$) appears at a latitude of approximately 27.5 ᵒN in the upper troposphere. This strong upward motion facilitates the diffusion and removal of the accumulated pollutants from the surface layer. According to Fig. 11e, the cluster with the largest frequency of 45% consists of the wet air parcels originating from Japan via the Yellow Sea. Only 5% of the trajectories originate from the Philippines and pass over the East China Sea. Overall, under the weather situation of Pattern 5, the transport of clean marine air masses and favorable diffusion conditions contribute to the good air quality over the YRD.

# Pattern 4

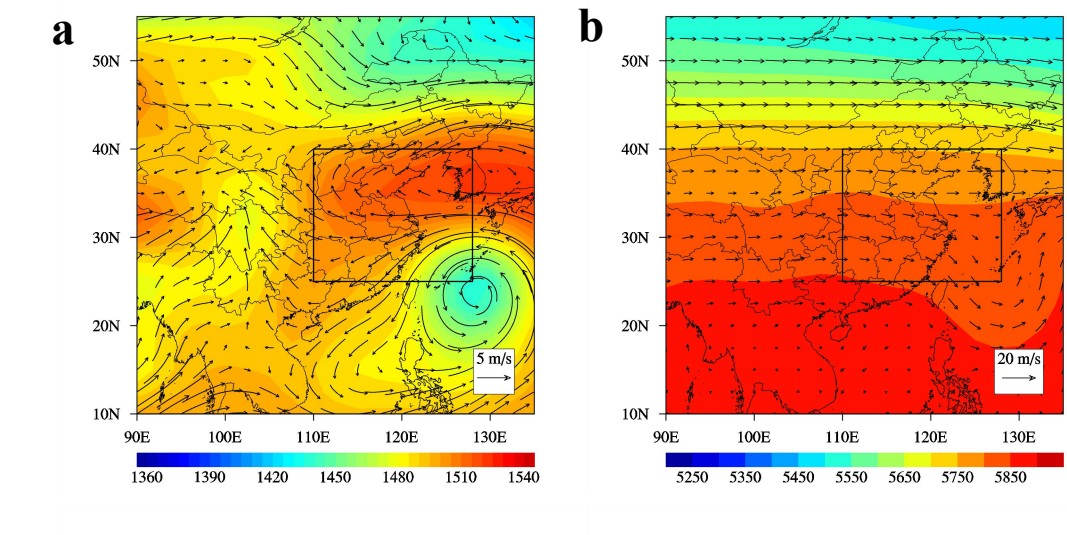

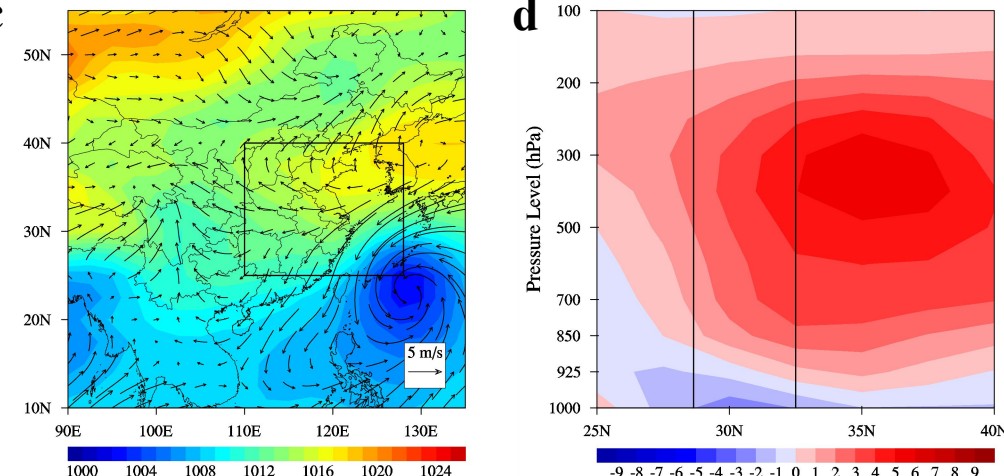

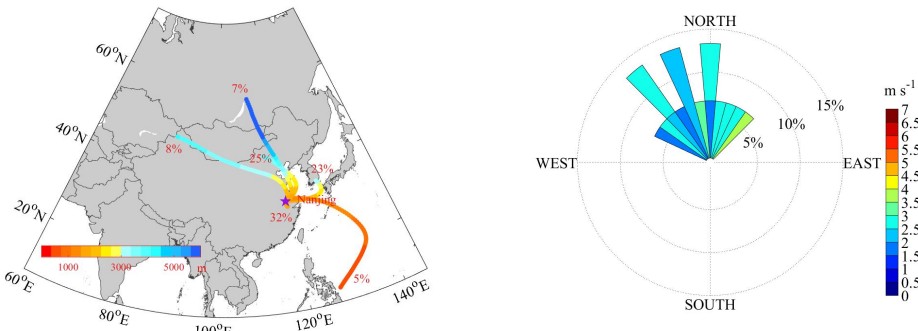



**Figure 10. As in Fig. 7, but for Pattern 4.**

# Pattern 5

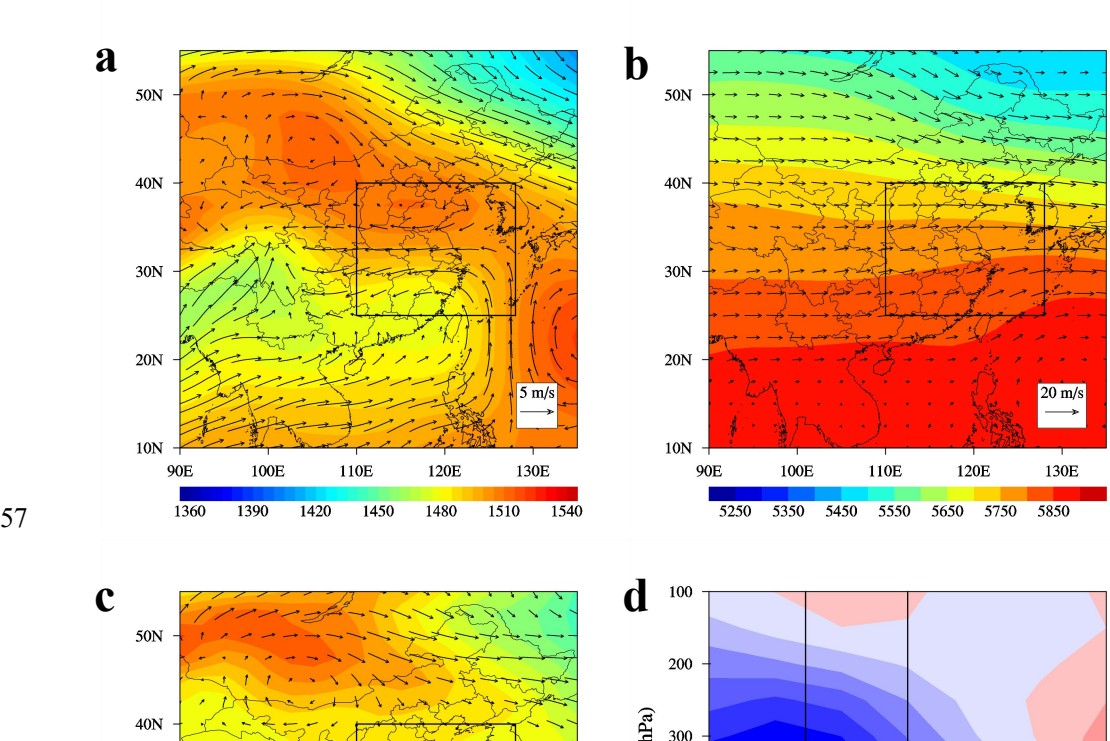



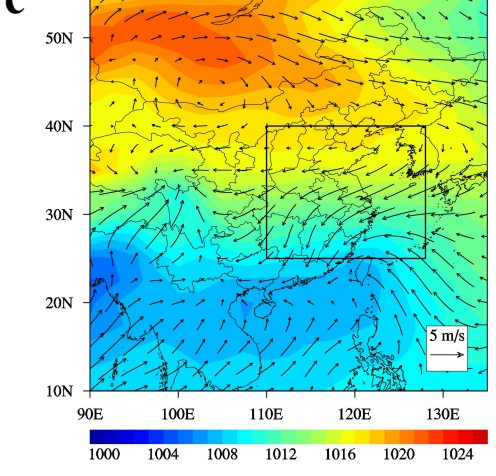

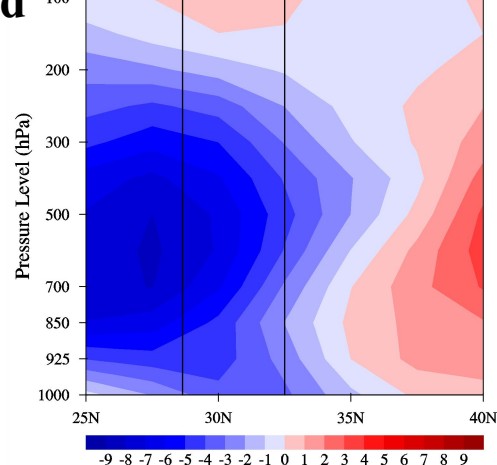

e                                      f

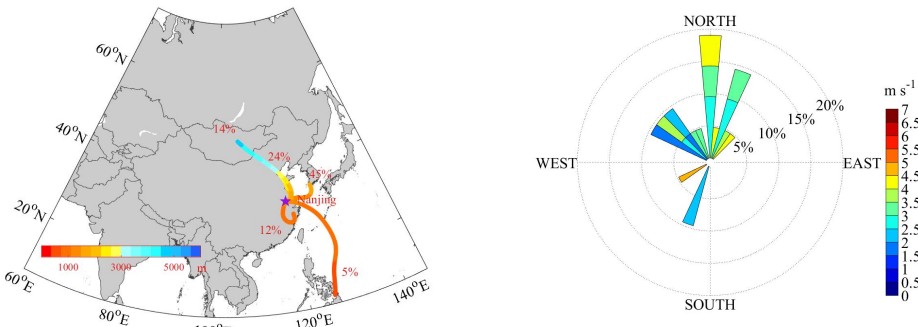


**Figure 11. As in Fig. 7, but for Pattern 5.**

To summarize, the weather situations for Patterns 1-5 are more or less affected by a
high-pressure system. However, the relative positions of the study area to the anti-cyclonic
circulation system have significant effects on the air quality of the YRD. These differences
determine the wind speed and wind direction, and the latter further determines whether the YRD is
influenced by the clean marine air masses. In both Pattern 1 and Pattern 3, the YRD is impacted
by the northwest air flows at the rear of the East Asian major trough, which transport abundant air
pollutants from other regions (such as the BTH and the SCB) to the YRD and cause severe particle
pollution (as well as high AOD values) in the YRD. In contrast, the weaker local surface wind in
Pattern 1 is extremely conducive to the local accumulation of pollutants. For this reason, Pattern 1
is 'the most polluted pattern', and it is responsible for most of the large-scale particle pollution
episodes over the YRD. Due to its stronger surface wind, Pattern 3 is 'the second-most polluted
pattern'. In Pattern 2, the polluted air masses mainly travel from the southern inland areas and
synchronously meet with the clean marine air masses in the YRD. To some extent, this weather
situation helps mitigate particle pollution in the YRD. In Pattern 4 and Pattern 5, the YRD is
directly influenced by air flows traveling from the ocean areas, and it is thus unlikely to be
polluted. Thus, Pattern 4 and Pattern 5 can be identified as 'the clean patterns'. These data suggest
that the clean marine air masses can substantially dilute the particle pollution over the YRD.

**4. Conclusions**
In this study, the spatial and temporal distributions of particle pollution in 16 YRD cities are
characterized from December 2013 to November 2014. Meanwhile, synoptic weather
classification is conducted to identify the dominant weather patterns over the YRD. The
meteorological fields and 72-h backward trajectories are analyzed to reveal the potential impacts
of weather systems on regional severe particle pollution episodes.

Observational records indicate that the concentrations of $PM_{2.5}$ and $PM_{10}$ decrease

progressively in the northwest-southeast direction. The pollution levels are comparatively higher
in Jiangsu Province and much lower in the southeast coastal area (i.e., Ningbo, Taizhou and
Zhoushan). The highest particle concentrations occur in Nanjing, where the concentrations of
$PM_{2.5}$ and $PM_{10}$ are 79 and 130 $\mu g \cdot m^{-3}$, respectively. The $PM_{2.5}/PM_{10}$ ratios are high in the YRD,
especially in winter. The seasonal mean $PM_{2.5}/PM_{10}$ ratios are 0.73 (winter), 0.61 (spring), 0.67
(summer) and 0.63 (autumn). These high $PM_{2.5}/PM_{10}$ ratios suggest that the $PM_{2.5}$ fraction is
extraordinarily dominant in the $PM_{10}$ mass in the YRD. In addition, high AOD values are also
found in the YRD, with an annual mean value of 0.71±0.57 and a maximum seasonal mean value
of 0.98±0.83 in summer. The diurnal cycles of the particle concentrations in most cities follow the
same pattern, reaching a morning peak from 8:00 to 12:00. There are three peaks in seasonal
variations (December, March, and May or June). The wintertime peak is closely related to
enhanced emissions during the heating season and poor meteorological conditions. Moreover, the
YRD suffers from $PM_{2.5}$ ($PM_{10}$) pollution on nearly 28.0% (13.1%) of the days of the year.
Continuous large-scale regional $PM_{2.5}$ pollution episodes occur much more frequently than $PM_{10}$
pollution episodes.

Based on the sums-of-squares technique, five typical synoptic weather patterns are

objectively identified in the YRD, including Pattern 1 (northwestly inland wind, which occurs on
47.7% of all days), Pattern 2 (southwestly, 20.0%), Pattern 3 (northly inland wind, 18.1%), Pattern
4 (cyclone-related, 4.1%) and Pattern 5 (oceanic circulation related, 5.8%).. Each pattern differs
from the other in respect to the relative position of the YRD to the main synoptic system (the
anti-cyclonic circulation system). This difference determines the wind speed and wind direction,
which play important roles in the air quality level of the YRD. In particular, the wind direction is
closely associated with determining whether the YRD is influenced by clean marine air masses. In
the patterns in which the YRD is located at the rear of the East Asian major trough at 850 hPa
(Pattern 1 and Pattern 3), strong northwest wind can easily transport air pollutants from other
polluted areas to the YRD, thus leading to serious particle pollution in the YRD. Due to the
high-pressure system, significant vertical downward motion is dominant above the YRD, resulting

in relatively stable weather conditions at the surface. With weak local surface wind, the worst polluted weather pattern (Pattern 1) features the highest regional mean $PM_{10}$ (116.5±66.9 μg·m$^{-3}$), $PM_{2.5}$ (75.9±49.9 μg·m$^{-3}$) and high AOD (0.74) values. Pattern 1 is also responsible for most of the large-scale regional $PM_{2.5}$ (70.4%) and $PM_{10}$ (78.3%) pollution episodes in the YRD. In Pattern 3, the strongest surface wind is conducive to the mitigation of pollution, thus resulting in the second-highest $PM_{10}$ (86.9±49.5 μg·m$^{-3}$) and $PM_{2.5}$ (59.1±37.3 μg·m$^{-3}$) values. In contrast, under the weather system of other synoptic patterns (especially Pattern 4 and Pattern 5), the clean marine air masses, which are transported via the east-southeast wind, play a crucial role in the mitigation of pollution over the YRD. Therefore, the YRD has a much smaller chance of being polluted.

In summary, the above results reveal that particle pollution in China is a thorny issue not only over a single city but also on a regional scale. This study can enhance our understanding of the features of particle pollution in East Asia. Meanwhile, these results also confirm that large-scale synoptic weather systems exert large impacts on regional particle pollution. Therefore, establishing potential links between different levels of particle pollution and predominant synoptic patterns can provide insight into formulating pollution control and mitigation strategies.

**5. Data availability**

The air quality monitoring records are available at http://106.37.208.233:20035. The meteorological data are available at http://www.nmc.cn. The MODIS/AOD records are available at https://ladsweb.nascom.nasa.gov/search/index.html. The NCEP reanalysis data are available at https://www.esrl.noaa.gov/psd/data/gridded/data.ncep.reanalysis2.pressure.html and http://ready.arl.noaa.gov/archives.php.

**Acknowledgments**

This work was supported by the National Natural Science Foundation of China (41475122, 91544230, 41621005), the National Key Research and Development Program of China (2016YFC0203303, 2016YFC0208504, 2017YFC0210106), and the open research fund of the Chongqing Meteorological Bureau (KFJJ-201607). The authors would like to thank the anonymous reviewers for their constructive and valuable comments on this manuscript.

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
