# Peer review of "Regional severe particle pollution and its association with"

_Atmospheric Chemistry and Physics, 2017_

## Referee Comment (RC1) · Anonymous Referee #1 · 28 Jun 2017

Particle pollution has been raised wide attention in the world, and is quite prominent in China. Synoptic system is identified as one of the significant causes. This paper studied the relationship between particle pollution and weather pattern in the Yangtze River Delta region of China. The work is meaningful. The manuscript is well organized. I suggest to publish the manuscript after addressing the comments and suggestions as below: 1) In Figure 2 and 3, it is better to mark the city name near each point. 2) The study is discussed the regional air pollution, but the used pollution data are mainly based on the surface monitoring records in 16cities. 16 points cannot well reveal the spatial characteristics of air pollution. So, it is better to use the MODIS/AOD data and add some more discussion based on them. The satellite information can help to show

the regional condition. 3) The analysis of transport processes of particle pollution is limited to the geopotential height fields and wind fields at 850 hPa. It is better to give a more comprehensive comparison between different layers, for example, at surface layer, 850 hPa layer, and 500 hPa layer, etc. 4) There are many grammar errors in this manuscript, including Lines 99-100, "a great deal of" are not a good choice of words. May be replaced by "a lot of researches"? Line 110, "focuses the pollution" should be replaced by "focuses on the pollution". Line 271, "the most importance source" should be replaced by "the most important source". Line 383, "occur for 14.3% of the days" may be revised as "occur in 14.3% of the days". Line 389, "are less frequently" should be replaced by "are less frequent". Lines 398-399, "Fig. 6 to 10" should be replaced by "Figs. 6 to 10". Many other errors are not pointed out here. Please improve the English of the manuscript with the aid of native speaker.

---

## Referee Comment (RC2) · Anonymous Referee #2 · 1 Jul 2017

In this manuscript, the regional characteristic of aerosol and its relation with synoptic weather patterns were discussed over the Yangtze River Delta region China. There are a lot of previous studies about PM10 and PM2.5 pollution in China. However, only a few of them have focused on the potential impacts of weather patterns on this kind of pollution. The results of this manuscript may be of great interests to the ACP audiences. Also, the study may be able to provide some useful views for the government on the air pollution control. Several comments and suggestions should be addressed before the publication of this paper. (1) Section 3.1.1 and 3.1.3. Apart from the in-situ monitoring particle concentration records, the aerosol optical depth data (monitored records, satellite observation, etc.) can be analyzed to deep the discussion on the

particle pollution in YRD. (2) Section 3.2.2, the author only mentioned and analyzed the geopotential height fields and wind fields at 850 hPa on the key date. The results may be quite different when it comes to the averaged condition of all days corresponding to each weather pattern. It's suggested to add the averaged geopotential height fields and revise the discussion. (3) Section 3.3.1, the occurrence frequencies of five weather patterns during the regional particle pollution episodes are not yet enough to conclude the relationship between them. It's suggested to add more detailed analysis for the monitoring data of particles (PM2.5 and PM10) and their precursors (such as SO2, NO2, etc.) at surface corresponding to each weather pattern. (4) Section 3.3.2, the wind speed and wind direction at surface are closely related to the transport processes. It's suggested to add the analysis of meteorological parameters from observational records corresponding to each weather pattern instead of NCEP reanalysis data. (5) The English should be polished. Some grammatical errors in this paper are listed as follows, Line 75, "Eastern Asian monsoon circulation" should be "East Asia monsoon circulation", "increasing aerosol loading" should be "increased aerosol loading". Line 110, "focuses the pollution" should be "focuses on the pollution". Line 271-272, "the most importance source" should be "the most important source". Line 577, "it also confirmed" should be "it was also confirmed". It is suggested to correct the errors with the aid of a professional language correcting company.
* * *

---

## Author Comment (AC1) · 31 Aug 2017

Particle pollution has been raised wide attention in the world, and is quite prominent in China. Synoptic system is identified as one of the significant causes. This paper studied the relationship between particle pollution and weather pattern in the Yangtze River Delta region of China. The work is meaningful. The manuscript is well organized. We appreciate the referee for the valuable and constructive reviews of our manuscript. We carefully revise the manuscript based on the following comments.

I suggest to publish the manuscript after addressing the comments and sugges-

tions as below:

1) In Figure 2 and 3, it is better to mark the city name near each point.

Response: Thanks for the constructive comment. The city names have been added in Figs. 2-3 in the new revised manuscript.

2) The study is discussed the regional air pollution, but the used pollution data are mainly based on the surface monitoring records in 16cities. 16 points cannot well reveal the spatial characteristics of air pollution. So, it is better to use the MODIS/AOD data and add some more discussion based on them. The satellite information can help to show the regional condition.

Response: Thanks for the constructive comment. In the new revised manuscript, the aerosol optical depth data from satellite observation (MODIS/AOD) are used to reveal the regional characteristics of aerosol pollution. The introduction of MODIS/AOD data has been added in Section 2.1. More discussion based on the AOD data has been added in Section 3.1.1, 3.1.2 and 3.3.1. These added data and discussion words can help us to understand the spatial distribution of aerosol in this region.

3) The analysis of transport processes of particle pollution is limited to the geopotential height fields and wind fields at 850 hPa. It is better to give a more comprehensive comparison between different layers, for example, at surface layer, 850 hPa layer, and 500 hPa layer, etc.

Response: Thanks for the constructive comment.

We have added the comparison of geopotential height fields and wind fields between different layers (500, 850 and 1000 hpa) in Section 3.3.2 of the new revised manuscript.

Meanwhile, in the new revised manuscript, we also removed Figs. 6-10 of the original manuscript, and replaced them with Figs. 7-11, which present the averaged

condition of all days for each weather pattern.

4) There are many grammar errors in this manuscript, including Lines 99-100, "a great deal of" are not a good choice of words. May be replaced by "a lot of researches"? Line 110, "focuses the pollution" should be replaced by "focuses on the pollution". Line 271, "the most importance source" should be replaced by "the most important source". Line 383, "occur for 14.3 percent of the days" may be revised as "occur in 14.3 percent of the days". Line 389, "are less frequently" should be replaced by "are less frequent". Lines 398-399, "Fig. 6 to 10" should be replaced by "Figs. 6 to 10". Many other errors are not pointed out here. Please improve the English of the manuscript with the aid of native speaker.

Response: Sorry for these grammatical errors in the original manuscript. The errors listed above are corrected as follows.

The words "a great deal of" on line 99-100 of the original manuscript are revised as "a lot of".

The words "focuses the pollution" on line 110 of the original manuscript are revised as "focuses on the pollution".

The words "the most importance source" on line 271 of the original manuscript are revised to "the most important source".

The words "occur for 14.3 percent of the days" on line 383 of the original manuscript are revised to "occur in 14.3 percent of the days".

The words "are less frequently" on line 389 of the original manuscript are revised to "are less frequent".

The words "Fig. 6 to 10" on line 398-399 of the original manuscript are revised to "Figs. 6 to 10".

Additionally, a professional language correcting company has helped to modify and improve the English in the new manuscript carefully.

---

## Author Response (AR1)

**Response to the comments of Referee #1:**

Particle pollution has been raised wide attention in the world, and is quite prominent in China. Synoptic system is identified as one of the significant causes. This paper studied the relationship between particle pollution and weather pattern in the Yangtze River Delta region of China. The work is meaningful. The manuscript is well organized.

We appreciate the referee for the valuable and constructive reviews of our manuscript. We carefully revise the manuscript based on the following comments.

I suggest to publish the manuscript after addressing the comments and suggestions as below:

1) In Figure 2 and 3, it is better to mark the city name near each point.

Response: Thanks for the constructive comment. The city names have been added in Figs. 2-3 in the new revised manuscript.

2) The study is discussed the regional air pollution, but the used pollution data are mainly based on the surface monitoring records in 16 cities. 16 points cannot well reveal the spatial characteristics of air pollution. So, it is better to use the MODIS/AOD data and add some more discussion based on them. The satellite information can help to show the regional condition.

Response: Thanks for the constructive comment. In the new revised manuscript, the aerosol optical depth data from satellite observation (MODIS/AOD) are used to reveal the regional characteristics of aerosol pollution. The introduction of MODIS/AOD data has been added in Section 2.1. More discussion based on the AOD data has been added in Section 3.1.1, 3.1.2 and 3.3.1. These added data and discussion words can help us to understand the spatial distribution of aerosol in this region.

3) The analysis of transport processes of particle pollution is limited to the geopotential height fields and wind fields at 850 hPa. It is better to give a more comprehensive comparison between different layers, for example, at surface layer, 850 hPa layer, and 500 hPa layer, etc.

Response: Thanks for the constructive comment.

We have added the comparison of geopotential height fields and wind fields between different layers (500, 850 and 1000 hpa) in Section 3.3.2 of the new revised manuscript.

Meanwhile, in the new revised manuscript, we also removed Figs. 6-10 of the original manuscript, and replaced them with Figs. 7-11, which present the averaged condition of all days for each weather pattern.

4) There are many grammar errors in this manuscript, including Lines 99-100, "a great deal of" are not a good choice of words. May be replaced by "a lot of researches"? Line 110, "focuses the pollution" should be replaced by "focuses on the pollution". Line 271, "the most importance source" should be replaced by "the most important source". Line 383, "occur for 14.3% of the days" may be revised as "occur in 14.3% of the days". Line 389, "are less frequently" should be replaced by "are less frequent". Lines 398-399, "Fig. 6 to 10" should be replaced by "Figs. 6 to 10". Many other errors are not pointed out here. Please improve the English of the manuscript with the aid of native speaker.

Response: Sorry for these grammatical errors in the original manuscript. The errors listed above are corrected as follows.

The words "a great deal of" on lines 99-100 of the original manuscript are revised as "many". Please see line 105 in the new revised manuscript.

The words "focuses the pollution" on line 110 of the original manuscript are revised as "focuses on the pollution". Please see line 115 in the new revised manuscript.

The words "the most importance source" on line 271 of the original manuscript are revised to "the most important source". Please see line 322 in the new revised manuscript.

The words "occur for 14.3% of the days" on line 383 of the original manuscript are revised to "occur on 14.3% of the days". Please see line 428 in the new revised manuscript.

The words "are less frequently" on line 389 of the original manuscript are revised to "are less frequent". Please see line 434 in the new revised manuscript.

The words "Fig. 6 to 10" on lines 398-399 of the original manuscript are revised to "Figs. 7 to 11". Please see lines 493-496 in the new revised manuscript.

Additionally, a professional language correcting company (Wiley Editing Services) has helped to modify and improve the English in the new manuscript carefully. Please see the revised manuscript with marks and the "language editing certificate".

**Response to the comments of Referee #2:**

In this manuscript, the regional characteristic of aerosol and its relation with synoptic weather patterns were discussed over the Yangtze River Delta region China. There are a lot of previous studies about PM10 and PM2.5 pollution in China. However, only a few of them have focused on the potential impacts of weather patterns on this kind of pollution. The results of this manuscript may be of great interests to the ACP audiences. Also, the study may be able to provide some useful views for the government on the air pollution control.

We would like to thank the referee for the valuable and affirmative comments of our manuscript. We carefully revise the manuscript based on the following comments.

Several comments and suggestions should be addressed before the publication of this paper.

(1) Section 3.1.1 and 3.1.3. Apart from the in-situ monitoring particle concentration records, the aerosol optical depth data (monitored records, satellite observation, etc.) can be analyzed to deep the discussion on the particle pollution in YRD.

Response: Thanks for the constructive comment. In the new revised manuscript, the aerosol optical depth data from satellite observation (MODIS/AOD) are used to reveal the regional characteristics of aerosol pollution and deep the discussion. The introduction of MODIS/AOD data has been added in Section 2.1. More discussion of AOD has been added in Section 3.1.1, 3.1.2 and 3.3.1. These added data and discussion words can help us to understand the spatial distribution of aerosol in this region.

(2) Section 3.2.2, the author only mentioned and analyzed the geopotential height fields and wind fields at 850 hPa on the key date. The results may be quite different when it comes to the averaged condition of all days corresponding to each weather pattern. It's suggested to add the averaged geopotential height fields and revise the discussion.

Response: Thanks for the constructive comment. In the new revised manuscript, we have removed Figs. 6-10 of the original manuscript, and replaced them with Figs. 7-11. Figs. 7-11 present the averaged condition of all days for each weather pattern. Meanwhile, according to the suggestion of Referee #1, we also added the comparison of geopotential height fields and wind fields between different layers (500, 850 and 1000 hpa) in Section 3.3.2 of the new revised manuscript.

(3) Section 3.3.1, the occurrence frequencies of five weather patterns during the regional particle pollution episodes are not yet enough to conclude the relationship between them. It's suggested to add more detailed analysis for the monitoring data of particles (PM2.5 and PM10) and their precursors (such as SO2, NO2, etc.) at surface corresponding to each weather pattern.

Response: Thanks for the constructive comment. More detailed analysis of the surface monitoring data of air pollutants (including $PM_{2.5}$, $PM_{10}$, $O_3$, $NO_2$, $SO_2$ and CO) for each weather pattern have been added in Section 3.3.1. Please see new Fig. 6 and the relevant discussion words in the new revised manuscript.

(4) Section 3.3.2, the wind speed and wind direction at surface are closely related to the transport processes. It's suggested to add the analysis of meteorological parameters from observational records corresponding to each weather pattern instead of NCEP reanalysis data.

Response: Thanks for the constructive comment. More detailed analyses for the surface monitoring data of meteorological parameters (wind speed, temperature, surface pressure and relative humidity) have been added in Section 3.3.1 of the new revised manuscript (new fig. 6 and the relevant discussion). In addition, the wind rose plots based on the daily data at the Nanjing site corresponding to each weather pattern from December 2013 to November 2014 are added in Figs. 7-11. The relevant discussion has also been added in Section 3.3.2 of the new revised manuscript. Besides, we also added the discussion of sea-level pressure field and wind field at 1000 hPa layer based on the NCEP reanalysis data, which can to some extent reflect the transport processes at the surface.

(5) The English should be polished. Some grammatical errors in this paper are listed as follows, Line 75, "Eastern Asian monsoon circulation" should be "East Asia monsoon circulation", "increasing aerosol loading" should be "increased aerosol loading". Line 110, "focuses the pollution" should be "focuses on the pollution". Line 271-272, "the most importance source" should be "the most important source". Line 577, "it also confirmed" should be "it was also confirmed". It is suggested to correct the errors with the aid of a professional language correcting company.

Response: Sorry for these grammatical errors in the original manuscript. The errors listed above are corrected as follows.

The words "Eastern Asian monsoon circulation" on line 75 of the original manuscript are revised as "East Asia monsoon circulation". The words "increasing aerosol loading" are revised as "increased aerosol loading". Please see lines 80-81 in the new revised manuscript.

The words "focuses the pollution" on line 110 of the original manuscript are revised as "focuses on the pollution". Please see line 115 in the new revised manuscript.

The words "the most importance source" on lines 271-272 of the original manuscript are revised as "the most important source". Please see line 322 in the new revised manuscript.

The words "it also confirmed" on line 577 of the original manuscript are revised as "these results also confirm ". Please see line 725 in the new revised manuscript.

Additionally, a professional language correcting company (Wiley Editing Services) has helped to modify and improve the English in the new manuscript carefully. Please see the revised manuscript with marks and the "language editing certificate".

[revised manuscript text omitted]

Given the pPrevious researchesstudies on of major climatic features in the YRD have demonstrated that, the southeast coastal area is dramatically affected by the land-sea breeze and marine air masses. The clean marine air masses are advantageous to the dilution and the diffusion of atmospheric pollutants, thus leading toproducing lighter air pollution. However, in the inland region, the clustered cities and the industrial districts tend to emit more pollutants, and thereby resulting in more the accumulation of accumulated more air pollutants around these cities.

[Figure]

**(a) Spatial distribution of AOD**

**(b) Temporal distribution of AOD**

[Figure]

[revised manuscript text omitted]
. , with the highest concentrations of PM$_{2.5}$ and PM$_{10}$ respectively being 79  and 130 μg·m$^{-3}$ in Nanjing.  The PM$_{2.5}$/PM$_{10}$ ratios are usually high , indicating that PM$_{2.5}$ is the overwhelmingly dominant particle pollutant in YRD. The wintertime peak of particle concentrations is tightly linked to the increased emissions in the heating season, as well as  the poor meteorological conditions. Secondly, based on NCEP reanalysis data, synoptic weather classification is conducted to reveal  the weather patterns that are easy to cause severe particle pollution in YRD. Five typical synoptic patterns are objectively identified, including the East Asian trough rear pattern, the depression inverted trough pattern, the transversal trough pattern, the high-pressure controlled pattern, and the northeast cold vortex pattern. Finally, synthetic analysis of meteorological fields and backward trajectories  are  applied to further clarify how these patterns impact particle concentrations. It is  demonstrated that air pollution is more or less influenced by  high-pressure system. The relative positions of YRD to the anti-cyclonic circulations  are quite significant to the air quality of YRD. YRD is largely influenced by polluted air masses from the northern and the southern inland areas when it is at the rear of the East Asian major trough. Significant downward motion of air masses results in stable weather conditions, and thereby  hinders the diffusion of air pollutants. Thus, the East Asian trough rear pattern is quite favorable for the accumulation of pollutants in YRD, and causes higher regional mean $PM_{10}$ ($116.5\pm66.9$ $\mu g \cdot m$-3), $PM_{2.5}$ ($75.9\pm49.9$ $\mu g \cdot m$-3) and AOD (0.74). Moreover, this pattern is also respnsible for the most occurrence of large-scale regional $PM_{2.5}$ (70.4%) and $PM_{10}$ (78.3%) pollution episodes. High wind speed and the clean marine air masses may play important roles in the mitigation of the pollution in 
[revised manuscript text omitted]

generally occur in  BTH, YRD , the Sichuan Basin (SCB) , and someand central and southern provinces in China (Hubei, Hunan and

Guangxi provinces).  AOD is mainly governed by fine particles in industrialized urban conditions (Kim et al., 2006), thus  these abovementioned areas should be suffering high columnar aerosol loading. In YRD, with the development of modern industrialization and urbanization, the contrasts of atmospheric pollution levels among the cities decrease gradually, and severe air pollution episodes tend to exhibit significant regional pollution characteristics.

Fig. 2b shows the temporal variation of regional averaged value of AOD in YRD

(covering 16 cities within the area of 25-40°N and 110-128°N)

.

The annual mean value AOD is 0.71±0.57.

The maximum seasonal value is 0.98±0.83 in summer, followed by 0.81±0.57 in winter, 0.59±0.24 in spring, and  0.48±0.35 in autumn .

Though the peak of particle concentrations occurs  in winter (as Fig. 3 and 5

shows), the above results demonstrate that the maximum regional mean AOD value occurs in summer, with the highest value of 1.60 in June.

The result is similar to that found by Kim et al. (2006). IIt is reported that the value of AOD is not only associated with the pollution levels of fine particles, but also strongly affected by other factors  (such as solar radiation, water vapor and etc.)

The maximum AOD value in hot seasons  should be ascribed to the combined effects of an increase of fine aerosol production (secondary aerosol formation by gas-to-particle conversion, hygroscopic growth of hydrophilic aerosols and biomass burning emissions) and humid weather (Kim et al., 2006).

Consequently, the aerosol optical depth data from satellite observation can reveal the spatial distribution of aerosols to some extent, but they cannot exactly reflect the pollution level and replace the concentration data.

Figs. 2c and 2d show the spatial distributions of annual mean particle concentrations in 16

typical cities over YRD from December 2013 to November 2014. Generally, the spatial distributions of PM$_{2.5}$ (Fig. 2c) and PM$_{10}$ (Fig. 2d) present a similar pattern as a whole. The annual mean PM$_{2.5}$ and PM$_{10}$ decrease progressively along the northwest-southeast direction, which means particle concentrations are comparatively high in the northwest inland areas and low in the southeast coastal areas. The pollution levels in most cities have a positive correlation with the proximity from the city to the sea. The farther the city is from the sea, the higher the concentrations are. The maximum particle concentrations occur in Nanjing, with the values of

79μg·m$^{-3}$ for PM$_{2.5}$ and 130 μg·m$^{-3}$ for PM$_{10}$. Given the previous researches on major climatic features in YRD, the southeast coastal area is dramatically affected by the land-sea breeze and marine air masses. The clean marine air masses are advantageous to the dilution and the diffusion of atmospheric pollutants, thus leading to lighter air pollution. However, in the inland region, the clustered cities and the industrial districts tend to emit more pollutants, and thereby result in more accumulated air pollutants around these cities.

[Figure]

[Figure]

[Figure]

**Figure 2.** Sspatial distribution of  annual mean AOD (at 550 nm wavelength) values over  YRD (a)

the temporal variation of regional averaged AOD value  over (28-33°N, 118-123°N) (b), the spatial distribution of annual mean PM₂.₅ concentrations (c), and the spatial distribution of annual mean

PM₁₀ concentrations (d). In (b), the gray line represents the daily value, the blue markers represent the

Fig. 3 illustrates the spatial distribution of  seasonal mean  PM2.5 in 16

cities over  YRD . The pattern in each season is similar to the annual mean pattern (Fig.

2a). The PM2.5 pollution levels are much higher in inland cities, and decrease along the northwest-southeast direction.  For the  seasonal variation, PM2.5

concentrations are highest in winter with the maximum value being up to 120 μg·m⁻³, decrease through the spring, and show the lowest values in summer and autumn. The difference between the PM2.5 concentration in summer and that in autumn is relatively small, _both_ with the maximum value lower than 60 μg·m⁻³ _in Nanjing_ and the minimum close to 20 μg·m⁻³ _in Zhoushan._

[Figure]

[Figure]

**Figure 3.  The s̶Sspatial distribution of  seasonal mean PM₂.₅  over  YRD  (a) winter, (b) spring, (c) summer, and (d) autumn. The acronyms for each city those in Figure 4.**

Table 1 quantitatively demonstrates the annual mean concentrations of PM₂.₅ and PM₁₀ in 16 cities over  YRD . It also shows that the particle pollution levels in inland cities are relatively higher. The concentrations of PM₂.₅ and PM₁₀ in 8 cities of Jiangsu province are all higher than 60 µg·m⁻³ (PM₂.₅) and 80 µg·m⁻³ (PM₁₀), respectively. However, the concentrations in the cities located in the coastal area (such as Ningbo, Taizhou and Zhoushan) are comparatively lower. Only the air quality of Zhoushan meets the national standard, which may be attributed to the fact that it is located on the island where the air is more likely influenced by the clean marine air masses.

To reveal the important role of PM₂.₅ in particle pollution, the ratios of PM₂.₅ concentration to

PM$_{10}$ concentration (PM$_{2.5}$/PM$_{10}$) are calculated over YRD. As listed in Table 1, the maximum annual mean value of the PM$_{2.5}$/PM$_{10}$ ratio is 0.72 in Shanghai, followed by Huzhou and Suzhou ( 0.71), implying that PM$_{2.5}$ fraction is overwhelmingly dominant of the PM$_{10}$

mass in these cities. The PM$_{2.5}$/PM$_{10}$ ratios in other cities are between 0.60 and 0.69, with the minimum value of 0.58 in Zhenjiang. These values are comparable to those in other cities like

Beijing (He et al., 2001), Shanghai (Wang et al., 2013), Taibei (Chen et al., 1999), and Hong

Kong (Ho et al., 2003), suggesting that the formation of PM$_{2.5}$ from gases is the most important source of particles in the cities of China. Table 1 also presents that the PM$_{2.5}$/PM$_{10}$ ratios in all cities show a distinct seasonal variation. It is remarkable that the values of PM$_{2.5}$/PM$_{10}$

are much higher in winter than in  other seasons, with the maximum value reaching 0.85 in

Shanghai and followed by 0.82 in Suzhou. The highest concentrations of PM$_{2.5}$ usually occur in winter (Fig. 3a) and high values of PM$_{2.5}$/PM$_{10}$ ratio also appear in the same season (Table 1), indicating that PM$_{2.5}$ poses a greater threat to human health in cold seasons that may be related to the heating activities. In summer, the values of PM$_{2.5}$/PM$_{10}$ in 16 cities  are medium, with the mean value of 0.67. The lowest ratios usually occur in spring and autumn, with the mean ratios of all cities being 0.61 (spring) and 0.63 (autumn). The minimum value occurs in the autumn of Yangzhou with the value of 0.51, followed by 0.52 in the spring of Nanjing and the autumn of Zhenjiang. The above discussion  on the spatial and temporal variations of

PM$_{2.5}$/PM$_{10}$ ratios also implies that particles originate from various kinds of sources and are variedly emitted.

**Table 1. Annual mean concentrations of PM$_{2.5}$ and PM$_{10}$, and the annual and seasonal mean values of PM$_{2.5}$/**

**PM$_{10}$ ratio in 16 cities over  YRD .**

| Cities | | PM$_{2.5}$ (µg·m$^{-3}$) | PM$_{10}$ (µg·m$^{-3}$) | PM$_{2.5}$/ PM$_{10}$ | | | | |
|---|---|---|---|---|---|---|---|---|
| | | | | Annual | Winter | Spring | Summer | Autumn |
| Shanghai | | 56 | 78 | 0.72 | 0.85 | 0.68 | 0.72 | 0.66 |
| Jiangsu Province | Nanjing | 79 | 130 | 0.61 | 0.64 | 0.52 | 0.70 | 0.60 |
| | Changzhou | 69 | 106 | 0.65 | 0.73 | 0.60 | 0.67 | 0.62 |
| | Nantong | 63 | 95 | 0.66 | 0.72 | 0.62 | 0.71 | 0.64 |
| | Suzhou | 67 | 94 | 0.71 | 0.82 | 0.68 | 0.71 | 0.67 |
| | Taizhoushi | 76 | 117 | 0.65 | 0.66 | 0.58 | 0.72 | 0.66 |
| | Wuxi | 75 | 114 | 0.66 | 0.73 | 0.59 | 0.67 | 0.62 |
| | Yangzhou | 68 | 114 | 0.60 | 0.69 | 0.58 | 0.59 | 0.51 |

| | | | | | | | |
|---|---|---|---|---|---|---|---|
| | Zhenjiang | 70 | 121 | 0.58 | 0.71 | 0.54 | 0.58 | 0.52 |
| | Hangzhou | 65 | 99 | 0.66 | 0.74 | 0.59 | 0.63 | 0.66 |
| | Huzhou | 68 | 96 | 0.71 | 0.78 | 0.66 | 0.68 | 0.69 |
| | Jiaxing | 58 | 84 | 0.69 | 0.75 | 0.65 | 0.68 | 0.69 |
| Zhejiang Province | Ningbo | 48 | 75 | 0.64 | 0.69 | 0.62 | 0.63 | 0.62 |
| | Shaoxing | 68 | 100 | 0.68 | 0.72 | 0.62 | 0.71 | 0.68 |
| | Taizhou | 50 | 75 | 0.67 | 0.69 | 0.66 | 0.66 | 0.65 |
| | Zhoushan | 31 | 50 | 0.63 | 0.66 | 0.62 | 0.66 | 0.55 |

**3.1.2 Temporal variations of particle pollution**

Fig. 4 shows the annual mean diurnal variation of $PM_{2.5}$ (Fig. 4a) and $PM_{10}$ (Fig. 4b) in 16 cities over YRD. Obviously, the diurnal cycles of particle concentrations in most cities follow the similar pattern. The $PM_{2.5}$ concentrations maintain comparably high values from 0:00 to 8:00 (local time). From then on, coinciding with more vehicle emission in rush hours, the concentrations go up rapidly from 8:00 to 12:00. After reaching the peak, $PM_{2.5}$ concentrations decrease and keep the low values until the sunset. During the nighttime, the pollutants get accumulated until the midnight, which might should be attributed to the more stable atmospheric stratification in the boundary layer. In comparison, there are two peaks in the diurnal cycles of $PM_{10}$ concentrations in several cities. The broad morning peak of $PM_{10}$ concentrations is more evident from 8:00 to 12:00, and the evening one occurs around 20:00. Besides, the diurnal change of particle concentrations in the southeast coastal area like Zhoushan is much smaller. As discussed in Section 3.1.1, the difference might be related to its special geographic location, low pollution level and less emission of precursors and low pollution level.

[revised manuscript text omitted]

As shown in Figs. 6a6f and Table 5, the highest concentrations of main air pollutants (except $O_3$) averaged for 16 cities in YRD are observed to be associated with Pattern 1. Since aerosols can reflect and absorb solar radiation (Kaufman et al, 2002), thus causing and thereby cause the decrease of the photochemical production of $O_3$ (Kaufman et al, 2002), the $O_3$ concentration is lowest for Pattern 1 (Fig. 6c). As above mentioned, Pattern 1 is most likely to occur in winter (30.5%) and spring (25.9%).

Therefore, the weather situation of this pattern generally features the weakest surface wind, the lowest humidity, the second highest surface pressure, and low temperature and relatively high surface pressure (only second to that for Pattern 3). All Tthese synoptic conditionsweather characteristics are conducive to an accumulation of particles and their precursors ($SO_2$, $NO_2$ and

CO). For Pattern 3, the concentrations of $PM_{10}$, $PM_{2.5}$ $NO_2$ and $SO_2$ are the second highest compared to other patterns, as well as the variability of all six air pollutants. This pattern features the highest surface pressure and much stronger surface wind. The temperature is lowest as Pattern

3 also tends to take place in winter (37.0%) and spring (30.1%). Under the weather situation of

Pattern 1 and Pattern 3, YRD is usuallyboth under the control of high-pressure system, and most likely to suffer heavyserious particle pollution. However, tThe strength of surface wind for different weather patterns plays a key role in the occurrence frequency of regional severe particle pollution episodes. With the weakest surface wind, making Pattern 1 is regarded as be'"the most polluted" pattern'". As for Pattern 2, the pollution levels of main pollutants are in the middle and slightly lower than those for Pattern 3. Due to the high occurrence frequency in summer (37.0%)

and spring (30.1), the weather condition of Pattern 2 is characterized as RH was found to be lowest, with relatively high temperature and, low pressure, withand the lowest RH. In contrast,

Pattern 4 and Pattern 5 are "'the clean least polluted" pattern'", with the concentrations of all pollutants concentrations being closely approximated and obviousdistinctly lower than other three patterns. The relatively high humidity, high temperature, strong wind speed (especially for Pattern

5) and much low surface pressure are favorable to the mitigationdiffusion of pollutants.

Furthermore, Figs. 6k to 6o display the spatial distribution of AOD over eastern China under different synoptic weather patterns. Thereinto, tThe regional mean values of AOD in YRD

(28-33°N, 118-123°N) corresponding to Pattern 1 to 5 are 0.74 for Pattern 1, 0.64 for Pattern 2,

0.81 for Pattern 3, 0.47 for Pattern 4 and 0.49 for Pattern 5 corresponding to Pattern 1 to 5, respectively. It can also be seenfound from Fig. 6 that AOD over YRD is highester for Pattern 3, followed by Pattern 1 and Pattern 2. For these three patterns, high AOD usually is observed coveringoccurs in large areas of China (BTH, YRD, SCB, anas well asd the provinces of Shanxi,

Shandong, Hubei, Hunan, Anhui and GuangxiEspecially, Thereinto, he highest

AOD are mainly found in northeastern China. For Pattern 4 and Pattern 5, high AOD is most concentrated in BTH and Shandong province while relatively low AOD is found in YRD. Since AOD is closely related to fine particles concentrations, it can be concluded that YRD is most heavily polluted under the weather situations of Pattern 1 and Pattern 3.

[Figure]

**Figure 6.** (a-j) Whisker-box plots for mean values of air pollutans concentrations and meterological parameters of 16 typical YRD cities . The edges of each box in (a-j) are the  25th and  75th percentilesthe band inside the box is the median, the diamond is the average, and the whiskers extend to the most extreme data. **(k-p)** Spatial distributions of AOD for the five synoptic weather patterns. P1, P2, P3, P4, and P5 represent Pattern 1, Pattern2, Pattern 3, Pattern 4, and Pattern 5, respectively.

**Table 5.** The average values of air pollut concentrations and meteorological factors  for 16 typical YRD cities  under the different synoptic weather patterns.

| Type | PM$_{10}$ | PM$_{2.5}$ | O$_3$ | NO$_2$ | SO$_2$ | CO | SO$_2$ | WS | T | P | RH |
|---|---|---|---|---|---|---|---|---|---|---|---|
| Pattern 1 | 116.5±66.9 | 75.9±49.9 | 57.7±27.3 | 46.9±19.2 | 29.3±17.1 | 1.08±0.41 | 29.3±17.1 | 1.84±0.67 | 15.8±7.8 | 1015.0±8.5 | 72.3±14.4 |
| Pattern 2 | 81.5±38.4 | 52.3±27.4 | 65.5±23.6 | 36.1±13.4 | 20.6±9.9 | 0.86±0.24 | 20.6±9.9 | 2.38±0.88 | 20.3±6.3 | 1011.2±6.7 | 79.8±10.2 |

| Pattern 3 | 86.9±49.5 | 59.1±37.3 | 58.5±25.5 | 35.1±15.5 | 23.3±15.9 | 0.96±0.35 | 23.3±15.9 | 2.59±0.87 | 13.4±8.2 | 1016.1±9.6 | 76.0±11.6 |
|---|---|---|---|---|---|---|---|---|---|---|---|
| Pattern 4 | 66.1±18.8 | 40.7±15.9 | 76.8±19.6 | 29.4±9.8 | 19.4±6.4 | 0.72±0.17 | 19.4±6.4 | 2.29±0.64 | 21.7±4.9 | 1011.8±7.0 | 75.4±5.8 |
| Pattern 5 | 58.7±31.3 | 37.4±22.5 | 61.1±20.6 | 29.1±11.1 | 17.8±8.4 | 0.77±0.22 | 17.8±8.4 | 2.63±0.93 | 19.4±8.0 | 1011.1±6.9 | 81.0±9.8 |

**3.3.2 The impact mechanism of synoptic weather patterns on  severe particle pollution**

Figs.  -110 present the meteorological fields and the backward trajectories under the weather situations of  the five synoptic weather patterns. The first two graphs of Figs. 76 10  illustrate the 850 hPa and 500 hPa geopotential height field and wind field, respectively. The third graphs display the sea level pressure field and 1000 hPa wind field  . The highlighted  boxes point out the essential area (YRD) that we focus on. The fourth graphs  demonstrate the height-latitude cross-sections of vertical velocity in the latitude (25-40°N), which is averaged from 110-128°E in the longitude. The bold black lines show the latitude range of 16 cities (28.6-32.5°N) over YRD. The positive wind speeds ($10^2$ Pa s$^{-1}$) indicate that there are vertical downward atmospheric motions, while the negative wind speeds represent the upward motion. Besides, it is well known that the atmospheric pollutant transport trajectories are deeply affected by synoptic systems. As shown in the last graphs  in Figs. 67 01, to reveal how the typical synoptic weather patterns influence the distribution of particles in YRD, the 72-h backward trajectories are calculated and then clustered. Given that Nanjing is the most polluted city in YRD as described in Section 3.1, the observational site in Nanjing (32°N, 118.8°E) is chosen for the terminus of the trajectories for each synoptic weather pattern.

As illustrated in Fig. 6a7a, Pattern 1 usually occurs when YRD is located at the rear of the East Asian major trough and under the control of a high-pressure ridge at 850 hPa. The center of the high-pressure system is on the northwestern Pacific Ocean. strongly affected by a low-pressure system at 850 hpa, namely the Aleutian Low. Meanwhile, northeastern China is strongly affected by a low-pressure system, namely the Aleutian Low.

 The strong horizontal northwest wind at the rear of the East Asian major trough can transport the pollutants from BTH (high AOD as shown in Fig. 6k) to YRD. At the same time,  the west and southwest wind at the rear of the high-pressure ridge can also transport the pollutants from central and southwestern China (such as SCB and Guangxi province) to YRD. The confluence of air flows may cause an  accumulation of pollutants in YRD. Accordingly, the atmospheric circulation at 500 hPa features a shallow through with west-northwest flow (Fig. 7b). The sea level pressure pattern is almost dominated by uniform pressure field, with relatively weak anti-cyclonic circulation over YRD (Fig. 6c). The above discussion can be further explained by the 72-h backward trajectories displayed in Fig. 6c. When YRD is under the control of Pattern 1, the air masses are mainly from northern China (44%), followed by central region (36%) and the north of YRD (19%). It suggests that the particle pollution is remarkably affected by the polluted air masses from BTH and central city clusters . Surface meteorological observation records also shown  that west-northwest, southwest  surface winds dominate in Nanjing (Fig. 7f), and high PM$_{2.5}$ is closely associated with the transport of polluted air masses in these wind direction.

In the vertical section (Fig. 6b), the relatively weak upward air flows dominate in the south of 30°N, while the clear downward air flows prevail in the north of 30°N. The largest descending velocity ( ~24×10$^{-2}$ Pa s$^{-1}$)  appears at the altitude of 500 hPa and in the latitude of  37.5°N. Downward motion dominates above YRD, which is in accordance with the 850 hPa circulation pattern represented by a high-pressure ridge. For this reason, the weather conditions are relatively stable near the surface and beneficial to the local accumulation of pollutants.  Overall, Pattern 1 represents a stable synoptic weather pattern, and this weather situation is extremely conductive to the built-up of atmospheric pollutants over YRD. This result is consistent with the finding of Zheng et al (2015b).

**Pattern 1**

[Figure]

[Figure]

[Figure]

[Figure]

[Figure]

**Figure** **7.** **Pattern 1****.** **(a) 850 hPa geopotential height field and wind field,** **(b) 500 hPa geopotential height field and wind field, (c) sea level pressure field and 1000 hPa wind field, (****d) height-latitude cross-sections of vertical velocity (unit: 10⁻² Pa/s) averaged from longitude of 110-128°E.**   **(****e) 72-h backward trajectory ending at the height of 1500 m, and (f) observation wind rose plots in Nanjing. In (a)-(c), the highlighted boxes point out the essential area (YRD) that we focus on. In (d), the black rectangular region represents the 16 cities in YRD (28.6-32.5°N). In (e), T****the purple marker indicates the location of Nanjing (32°N, 118.8°E). The data is averaged for all days corresponding to Pattern 1.**

As for Pattern 2, a  low-pressure center (the Southeast Vortex) aris centered in SCB, the East China Sea is influenced by a high-pressure system, and a depression inverted trough extends and covers the YRD region in latitude at 850 hPa (Fig. 8a). Consequently, in YRD, the strong southwest air flows from southern China meet with the southeast air flows from the East China Sea. After the convergence of air masses, they jointly transport pollutants northwestward. While at surface (Fig. 8c), the study domain is located at the bottom of a high-pressure system  and impacted by strong southeast wind . In the middle troposphere (Fig. 8b), the sparse isopleths indicate small geopotential height gradient, while the shallow ridge causes westerly flows. Fig. 8c also illustrates these air pollutant transport paths. For the days when Pattern 2 dominates, about 42% of the air masses are from the southwest and the south of China, and 15% are from the East China Sea.

to the

The air masses from the East China Sea are very important, because the clean marine air masses may dilute the particle concentrations in YRD. Besides, there are nearly 43% air masses originating from the local sources of YRD, which may be related to the short-range transport in the northwest direction. This is also in accordance with the dominant northwest surface wind in Nanjing (Fig. f). When it comes to the vertical structure (Fig.

8c), Pattern 2 is obviously different from Pattern 1, as the upward air flows dominate in the south of 37.5°N.

The largest updrafts zone (<15×10⁻² Pa s⁻¹) appears above YRD and between the altitude of 700 hPa and 500 hPa. The vertical velocity close to surface is relatively weaker compared to that at higher levels over YRD. Meantime, there is stronger upward motion near surface in the latitude of 37.5°N, with weak downward motion above the 700 hPa layer.

This difference suggests that atmospheric pollutants in YRD are horizontally transported northwestward to higher latitude, and vertically transport upward to high layers. Therefore, despite the transport of abundant pollutants to YRD via southwest air flows and short-range northwest transport of polluted air masses, the strong surface southeast wind and upward motion under the weather situation of

Pattern 2 determine that there is much slighter particle pollution over YRD compared to Pattern 1.

## Pattern 2

[Figure]

[Figure]

**Figure 87.–_As in Fig. 76 but for Pattern 2.**

For Pattern 3, it tends to occur in winter (36.4%, as displayed in Table 3). Under this circumstance, YRD is mainly controlled by a high-pressure system centered in central China (Fig. 98a). Meanwhile,

northeastern China is under the steering influence of the northwest air flows at the rear of the East Asian major trough, with . the its trough axis appearing along the eastern coastline of

China.

Affected by the strong notherwest winds coming from

Northorthern China, the polluted  air masses from BTH are easily transported to YRD. At the higher layer of 500 hPa (Fig. 98b), the and wind fieldcirculation structure pattern are similar to those for  of Pattern 1. A trough appears in the upper atmosphere, resulting in relatively strong west-northwest flows. The dense isopleths indicate large geopotential height gradient and strong downward flows. While at the surface layer (Fig. 98c), the strong northerly wind is also evident, and YRD is located at the bottom of a high-pressure system centered in the remote Mongolian region.  in the north of transversal trough are slowed down. The above discussion is further proved by the results from back trajectory calculations. As suggested in Fig. 8e98e, most air masses in clusters are from the Loess Plateau, with the percentage of 31%. The transport path of this cluster is relatively short, which might be attributed to the weakened strong anti-cyclonic circulationnorthwest wind. Due to the strong northerly windFor this reason, tThe long-range transport of air masses from remote Mongolia and northernnorth China also accounts for 22% and 18% of all trajectories, respectively. Besides, the local transport of air masses from the southeast coastal area in YRD accounts for 26%, which is associated with the northeast air flows. The marine air masses cluster originates from western Pacific via the Yellow Sea accounts for 4%. They both bring the clean marine air masses to YRD, which is somewhat beneficial to the mitigation of particle pollution in YRD.? For the vertical structure (Fig. 8b98d), the distribution of vertical velocitflow fieldy below the altitude of 300 hPa is similar to that of Pattern 1, whereas the vertical wind is slightly slower stronger for the weather systems in Pattern 3. UndeDue tor the steering influence of the high-pressure system, it is observed that The evident downward air flows dominate in the north of around 28°N (including YRD) below the altitude of 300 hPa. The largest descending velocity (~$9\times10^{-2}$ Pa s$^{-1}$) also appears at the altitude of 500 hPa, covering the latitude of 35-40°N.Thus, influenced by the downdrafts in higher latitudes and horizontal northeast air flows, more clean marine air masses may be transported to YRD. Due to the fact that YRD is under the steering influence of the high-pressure system, downward motion dominates above YRD as well, the same to that of Pattern 1.However, in despite of  the higher surface pressure (Figs. 6i and 98c) and stronger downward motion (Fig. 98d), the surface wind is much stronger for Pattern 3 as well (as displayed in Figs. 6g, 98a and 98c), which alleviating es the problems of air pollution resulting in much slighter air pollution over YRD compared tothan that of Pattern 1. In all, under the weather situation of  Pattern 3, the strong northwesterly wind in the front of the high-pressure system usually lead to the transport of polluted air masses from BTH to YRD may cause particle pollution in YRD when the north

. Nevertheless, the strong surface wind is conducive to the mitigation of pollutants

, which plays a significant role in the level of air pollution over YRD

.

**Pattern 3**

[Figure]

**a**

**b**

**c**

**d**

**e**

**fe**

[Figure]

**Figure 98. As in Fig. 76 but for** **Pattern 3**.

    With respect to Pattern 4, on both  surface and 850 hPa level, the study domain is

  under the control of a high-pressure system  (Figs. 109a and 109c). The center of the

  high-pressure system is located on the Sea of Japan, while a cyclonic circulation occurs over the

  Philippine Sea. The anti-cyclonic circulation prevails over YRD and horizontally brings the clean marine air masses to the land. Meanwhile, the sparse isopleths represent  smallweak geopotential height gradient in the middle troposphere, accompanied by muchrather weaker west wind compared to  other patterns (Fig. 109b). Accordingly, influenced by the high-pressure system, the downward atmospheric motion dominates in the vertical direction obviously (Fig. 9b109d). The strongest downward motion ($\sim6\times10^{-2}$ Pa s$^{-1}$) appears between the altitude of 300 hPa and 500 hPa and at the latitude of 35$^{\circ}$N. The weak updrafts near the surface may be related to the regional thermodynamic circulation. As shown in Fig. 109ee, the cluster with the largest frequency of 32% stands for the local transport of air masses from southern adjacent areas in YRD. Additionally, the air masses from northern China via Bohai Bay (25%), from Japan via the Yellow Sea (23%), and from the Philippines via the East China Sea (5%) are also representative. These clusters passing over the ocean areas totally account for more than 50% of all trajectories. Therefore, under this weather situation, it is confirmed that the dilution effects of clean marine air masses play great roles in the particle pollution over YRD.

Pattern 5 features one of the most complex circulation situation at 850 hPa (Fig. 11a). YRD is located between the bottom of the northern high-pressure system and the top of the southern weak low-pressure system. For this reason, the horizontal strong east wind prevails and easily carries clean marine air masses from the East China Sea to YRD. The corresponding circulation structure at the surface layer is similar to that at 850 hPa layer (Fig. 11c), while the east-northeast flows prevails over the study domain. In the upper troposphere, a ridge appears in the east due to the tropical cyclonic system, thus leading to the west-southwest flows over the region. Owing to the above-mentioned two opposite pressure systems (Fig. 11a), strong upward air flows are dominant in the south of the latitude of 35 $^{\circ}$N, while the downward motion is obvious in the north (Fig. 11d). The largest ascending velocity ($\sim -9\times10^{-2}$ Pa s$^{-1}$) appears in the latitude of around 27.5 $^{\circ}$N in the upper troposphere. The strong upward motion facilitates the diffusion and removal of the accumulated pollutants from the surface layer. According to Fig. 11e, the cluster with the largest frequency of 45% consists of the wet air parcels from Japan via the Yellow Sea. Only 5% of the trajectories originates from the Philippines and pass over the East China Sea. On the whole, under the weather situation for Pattern 5, the transport of clean marine air masses and favorable diffusion condition contribute to the good air quality over YRD.

**Pattern 4**

[Figure]

**c**

**a**

**d**

**e**

**f**

[Figure]

(c)

**Figure 9. As in Fig. 6 but for Pattern 4.**

 Pattern 5 features  the most complex circulation situation  level (Fig.

0a).  The northeastern China is controlled by a cold eddy system. The central China is impacted by a high-pressure ridge. A strong tropical low-pressure system is located around

Luzon. At this time, YRD is located in  the south  of the northern high-pressure system and the north  of the  weak low-pressure system.

The horizontal  southeast wind prevails and  carries clean marine air masses from the East China Sea to YRD.

9while the east-northeast flows prevails over the study domain. In the upper troposphere, a shallow ridge appears in the east due to the tropical cyclonic system, thus causing to the west-southwest flows over the region. Owing to the above-mentioned two opposite pressure systems (Fig. 11a), strong   At the same time, upward air flows are dominant in the south of the latitude of 35 °N, while the downward motion is obvious in the north and comparatively weak (>-3×10⁻² Pa s⁻¹) in the lower troposphere (Fig. 11db). The largest ascending velocity (~ -9×10⁻² Pa s⁻¹) appears in the latitude of around 27.5 °N in the upper

c,

under ystems in  Pattern 4 and 5 are both mainly influenced by the and favorable diffusion condition contribute to the good air quality over YRD.

and largely beneficial to the diffusion of the pollutants.

<h2 style="text-align:center">Pattern 5</h2>

[Figure]

[Figure]

[Figure]

**Figure 110. As in Fig. 6 but for  Pattern 5.**

803  To sum up, the weather situation for Pattern 1-5 are more or less affected by a high-pressure

804 system. However, the relative positions of the study domain to the anti-cyclonic circulation system

805 are quite significant to the air quality of YRD. The differences determine the wind speed and wind

806 direction, and the latter further determine whether YRD is influenced by the clean marine air masses. For Patten 1 and Pattern 3, –YRD are both impacted by the northwest air flows at the rear of the East Asian major trough, which transport abundant air pollutants from BTHother regions (such as BTH and SCB) to YRD and cause heavysevere particle pollution (high AOD value as well) in YRD. In contrast, the weaker local surface wind for Pattern 1 is extremely conducive to the local accumulation of pollutants.under the influence of weather system of Pattern 1, the particle pollution in YRD is largely affected by the transport of pollutants from the south and north inland regions of China. This weather situation is extremely not favorable to the diffusion of air pollutants, For this reason, Pattern 1 is 'the most polluted pattern', and responsible for the most large-scale particle pollution episodes over YRD. Owning to the stronger surface wind, Pattern 3 is 'the second most polluted pattern'. As for Pattern 2 and Pattern 3, the polluted air masses mainly travel from southern inland areas, and synchronously meet with the clean marine air masses in YRD. To some extent, Tthis weather situation is helpful to the mitigation of particle pollution in YRD to some extent, and this pattern can also be regarded as 'the polluted pattern'. With respect to Pattern 4 and Pattern – and Pattern 5, YRD is directly influenced by the air flows traveling from the ocean areas, and has little chance of being polluted. Thus, Pattern 4 and Pattern -5 can be identified as 'the clean pattern'. It suggests that the clean marine air masses have great dilution impacts on the particle pollution over YRD.

**4. Conclusions**

In this study, the spatial and temporal distributions of particle pollution in 16 YRD cities are characterized from December 2013 to November 2014. Meanwhile, the synoptic weather classification is conducted to identify the dominant weather patterns over YRD. The meteorological fields and 72-h backward trajectories are analyzed to reveal the potential impacts of weather systems on the regional severe particle pollution.

From the observational records, it is shown that the concentrations of $PM_{2.5}$ and $PM_{10}$ decrease progressively along the northwest-southeast direction. The pollution levels are comparatively high in the Jiangsu Province and much lower in the southeast coastal area (Ningbo, Taizhou and Zhoushan). The highest particle concentration occurs in Nanjing, with the concentrations of $PM_{2.5}$ and $PM_{10}$ being 79 $\mu g \cdot m^{-3}$ and 130 $\mu g \cdot m^{-3}$, respectively. The $PM_{2.5}/PM_{10}$ ratios are high in YRD, especially in winter. The seasonal mean $PM_{2.5}/PM_{10}$ ratios are 0.73

 (winter), 0.61 (spring), 0.67 (summer) and 0.63 (autumn), respectively. These high $PM_{2.5}/PM_{10}$

ratios suggest that the $PM_{2.5}$ fraction is extraordinarily dominant in the $PM_{10}$ mass in YRD.

Besides, high AOD is also found in YRD, with the annual mean value of 0.71±0.57 and the maximum  seasonal mean value  of 0.98±0.83 in summer . The diurnal cycles of particle concentrations in most cities follow the same pattern, with a morning peak from 8:00 to 12:00. There are three peaks in seasonal variations (December, March, and May or June). The wintertime peak is closely related to the enhanced emissions in the heating season and poor meteorological conditions. Moreover, YRD  suffers from the $PM_{2.5}$ ($PM_{10}$)

pollution in nearly 28.0% (13.1%) days of the year. The continuous large-scale regional $PM_{2.5}$

pollution episodes occur much more frequently than the $PM_{10}$ pollution episodes.

Based on the sums-of-squares technique, five typical synoptic weather patterns are objectively  identified in YRD, including the East Asia major trough rear pattern (Pattern

1, occurs 47.7% of all days), the depression inverted trough pattern (Pattern 2, 20.0%), the transversal trough pattern (Pattern 3, 18.1%), the high-pressure controlled pattern (Pattern 4, 4.1%)

and the northeast cold vortex pattern (Pattern 5, 5.8%). Each pattern differs from the other in respect to the relative position of YRD to the main synoptic system (anti-cyclonic circulation system). The difference determines the wind speed and wind direction, which play an important role in the air quality level of YRD. Especially, the wind direction is closely associated with the situation whether YRD is influenced by clean marine air masses.

 Under the patterns when YRD is at the rear of the East Asian major trough at 850 hPa (Pattern 1 and Pattern 3), the strong northwest wind can easily transport air pollutants from other polluted areas to YRD, leading to serious particle pollution in YRD. Due to the high-pressure system, significant vertical downward motion dominates above YRD, resulting in relatively stable weather conditions at the surface. With weak local surface wind, the worst polluted weather pattern (Pattern 1) features the highest regional mean $PM_{10}$ (116.5±66.9 μg·m$^{-3}$), $PM_{2.5}$ (75.9±49.9 μg·m$^{-3}$) and high AOD (0.74). Pattern is also responsible for humidity and temperature). ~~The analysis of meteorological field also indicates that the strong horizontal northwest wind hinders the vertical outward transport of pollutants. Thus, this weather situation is extremely unfavorable for the diffusion of the pollutants, leading to the highest PM$_{10}$ (116.5±66.9 μg·m$^{-3}$), PM$_{2.5}$ (75.9±49.9 μg·m$^{-3}$) and high AOD (0.74). For this reason, Pattern 1 can be regarded as 'the most polluted pattern', and responsible forand contributes most to the occurrence of large-scalethe the strong northerly wind usually leads to the transport of polluted air masses from BTH to YRD, while the high-pressure system causes dominant downward motion over the region. hHighest AOD (0.81) is observed in YRD under this pattern. However,.Thus, Pattern 3 is supposed to be "the polluted pattern", withthe(,sUnder 
[revised manuscript text omitted]
 the YRD, only focuseds on the pollution in October, and is was mainly on basisbased of on satellite aerosol optical depth (AOD) data. Ground-based monitoring particle concentration data can better represent the status of particle pollution in the urban atmosphere of the YRD. Thus, to better understand the relationship between the pollution in the planetary boundary layer and the synoptic weather patterns over the YRD, further study studies should be conducted based on surface monitoring the data collected over a time period of at least over a one year from the surface monitoring inin the YRD.

This work attempts to enhance the our understanding of particle pollution in the YRD and, and provides the scientific knowledge for about the association of regional severe particle pollution and synoptic weather patterns. Firstly, First, we analyze the spatial and temporal distribution of PM$_{10}$, PM$_{2.5}$ and AOD in the YRD from December 2013 to November 2014 , aimed to illustrate the characteristics of particle pollution over the this region. Secondly, Second, synoptic weather classification is conducted to reveal the weather patterns related to heavy pollution. Finally, the synthetic analyseis of meteorological fields and backward trajectories are used to further clarify the impact mechanism. In this paper, Section 2 describes the observed data, the synoptic weather classification method and the trajectory model. Section 3 presents our main findings, including the a detailed analysis of the characteristics of particle pollution in the YRD, the synoptic weather patterns affecting the this pollution, and the mechanism how by which weather systems impact the pollution. In the endFinally, a brief summary is addressed 
[revised manuscript text omitted]

[Figure]

**LANGUAGE EDITING**

**CERTIFICATE**

This document certifies that the manuscript listed below was edited for proper English language, grammar, punctuation, spelling, and overall style by one or more of the highly qualified native English speaking editors at Wiley Editing Services.

**Manuscript title:**

Regional severe particle pollution and its association with synoptic weather patterns in the Yangtze River Delta region, China

**Authors:**

Lei Shu, Min Xie, Da Gao, Tijian Wang, Dexian Fang, Qian Liu, Anning Huang, Liwen Peng

**Date Issued:**

September 3, 2017

**Certificate Verification Key:**

02A8-9799-C450-AD88-F648

This certificate may be verified at https://secure.wileyeditingservices.com/certificate. This document certifies that the manuscript listed above was edited for proper English language, grammar, punctuation, spelling, and overall style. Neither the research content nor the authors' intentions were altered in any way during the editing process. Documents receiving this certification should be English-ready for publication; however, the author has the ability to accept or reject our suggestions and changes. If you have any questions or concerns about this document or certification, please contact help@wileyeditingservices.com.

[Figure]

Wiley Publishing Services is a service of Wiley Publishing. Wiley's Scientific, Technical, Medical, and Scholarly (STMS) business serves the world's research and scholarly communities, and is the largest publisher for professional and scholarly societies. Wiley is committed to providing high quality services for researchers. To find out more about Wiley Editing Services, visit wileyeditingservices.com. To learn more about our other author services provided by Wiley Publishing, visit authorservices.wiley.com.